# Characterization and bioactivity potential of marine sponges (*Biemna fistulosa, Callyspongia diffusa,* and *Haliclona fascigera*) from Kenyan coastal waters

Teresia Nyambura Wacira[1,2]*, Huxley Mae Makonde[1], Joseph Nyingi Kamau[3], Christopher Mulanda Aura[2], Cromwell Mwiti Kibiti[1]

1 Department of Pure and Applied Sciences, Technical University of Mombasa, Mombasa, Kenya,
2 Kenya Marine and Fisheries Research Institute, Freshwater Research Center, Kisumu, Kenya, 3 Kenya Marine and Fisheries Research Institute, Mombasa, Kenya

* tnyambura@kmfri.go.ke

## Abstract

Sponges have been reported as a rich source of bioactive compounds, which could potentially be developed into lead compounds for pharmaceutical use. The present study aimed to identify sponges and assess the biological activity of their extracts against human disease-causing organisms, including *Escherichia coli*, *Pseudomonas aeruginosa*, *Staphylococcus aureus,* and *Candida albicans*. Morphological characterization and DNA barcoding of the cytochrome c oxidase subunit I (COI) gene characterized three sponge species (*Biemna fistulosa*, *Callyspongia diffusa* and *Haliclona fascigera*). The Kirby-Bauer test assessed the antimicrobial activity of the extracts, and the inhibition zone diameters (IZD) were measured. The extracts were further subjected to minimum inhibitory concentration (MIC) and minimum bactericidal concentration (MBC) tests to determine the antibiotic susceptibility. The Gas Chromatography-Mass Spectrometry (GC-MS) was used to identify and quantify the organic compounds in the sponges' extracts. The methanolic extract of *B. fistulosa* (28.00 ± 3.5 mm) and *H. fascigera*. (28.33 ± 3.8 mm) exhibited a broad spectrum of antibacterial activity against *E. coli,* surpassing the positive control (27.67 ± 0.9 mm). The inhibitory activity of ethyl acetate extract of the *C. diffusa* (29.33 ± 2.4 mm) against *P. aeruginosa* was observed to be higher compared to the standard antibiotic streptomycin (26.67 ± 0.7 mm). The methanolic extract of *H. fascigera* demonstrated the lowest MIC (0.53 ± 0.0 mg mL$^{-1}$) compared to the streptomycin drug (1.36 ± 0.0 mg mL$^{-1}$), and showed an MBC of 1.25 mg mL$^{-1}$ against *E. coli*. The GC-MS chromatogram data analysis identified 114 distinct compounds categorized into 39 classes across three sponge extracts: 11.4% of these compounds demonstrated documented antimicrobial activity against human pathogens. This study corroborates sponges as a promising source of bioactive compounds, which are valuable leads for drug

**Data availability statement:** All relevant data are within the manuscript and its Supporting Information files.

**Funding:** We sincerely thank the Kenya Marine and Fisheries Research Institute (KMFRI) for their invaluable support in partially funding for the research (GOK-PC Target C82 39-1). The funders had no role in study design, data collection and analysis, decision to publish, or preparation of the manuscript.

**Competing interests:** The authors have declared that no competing interests exist.

discovery and development. Future research must explore their mechanisms, molecular-level toxicity, and lead optimization to enhance drug development.

## Introduction

Despite a record dating back at least the entire Phanerozoic, approximately 600 million years [1], the magnitude of aquatic marine sponge biodiversity is unknown [2]. As of March 2025, the World Porifera Database (WPD) [3] recognizes approximately 9,722 accepted living sponge species and 60 extinct species. In 1950, Bergmann and Feeney discovered marine sponge natural products by extracting novel spongothymidine and spongouridine nucleosides from *Tectitethya crypta*, formerly known as *Cryptotheca crypta* [4]. Among the three classes of sponges, class *Demospongiae* and orders *Halichondrida, Poecilosclerida*, and *Dictyoceratida* are the significant sources of bioactive compounds compared to *Hexactinellida* and *Calcarea* [5].

Recent scholarly investigations underscore the considerable potential of marine sponges as a source for novel antimicrobial agents. Notably, a study conducted on sponges from Saint Thomas, U.S. Virgin Islands, demonstrated significant antibacterial activity against several pathogenic microorganisms, including *Escherichia coli, Staphylococcus aureus*, and *Neisseria gonorrhoeae* [6]. More than 200 new compounds have been isolated from marine sponges, accounting for approximately 23% of approved marine-derived pharmaceuticals [7].

Dragmacidin G, a bioactive alkaloid extracted from sponges of the genera *Spongosorites* and *Lipastrotheya*, exhibited a broad spectrum of biological activity, including the inhibition of *Mycobacterium tuberculosis*, *Plasmodium falciparum* and methicillin-resistant *Staphylococcus aureus* (MRSA) [8]. Manzamine A, derived from Indo-Pacific sponges, manifests broad-spectrum antibacterial activity against both Gram-positive and Gram-negative bacterial strains [9]. Similarly, Aeroplysinin-1, extracted from *Aplysina aerophoba*, has revealed significant antibacterial properties against various pathogenic strains [10]. Additionally, compounds such as Discorhabdin G and Ageliferin from sponges have been recognized for their ability to prevent bacterial biofilm formation, a critical factor that contributes to the persistence and resilience of infections. Furthermore, the pharmacological potential of marine sponges transcends antibacterial activity, as they are also abundant sources of antifungal and antiviral agents [11]. Avarol, a hydroquinone derivative isolated from the sponge *Dysidea avara*, has antifungal activity against various pathogenic fungi [12]. Mycalamide A, obtained from *Mycale* sp., is renowned for its antiviral activity, particularly against herpes simplex virus and poliovirus type 1 [13].

Despite the significant advancements made in the research of marine sponges for the production of bioactive compounds, considerable gaps remain in our understanding of marine sponges and their bioactive compounds [14]. Furthermore, while various sponge-derived compounds are recognized for their antibacterial and antifungal activities, there is inadequate literature and comprehensive data on the identification and bioactivity studies of marine sponges from the Kenyan coastal waters [15].

Morphological identification of marine sponges is constrained by phenotypic plasticity, environmental variability, and the presence of cryptic species, while DNA barcoding has enhanced taxonomic resolution but remains hindered by incomplete molecular databases for Kenyan coastal sponges [16]. Extraction methodologies significantly influence compound recovery, with methanolic extracts facilitating the isolation of polar bioactive molecules such as alkaloids and flavonoids, albeit with potential thermolabile degradation, ethyl acetate extracts exhibiting superior antimicrobial activity yet limited in their ability to retain highly polar compounds, and dichloromethane extracts preferentially isolating non-polar sterols and terpenoids but demonstrating reduced efficacy against hydrophilic pathogens [17].

Although GC-MS analysis has identified numerous distinct compounds, their pharmacological properties, toxicity profiles, and mechanisms of action remain largely unexplored, necessitating further investigation to elucidate their bioactive potential, while oceanographic factors such as seasonal monsoons, nutrient upwelling, and coral reef ecosystems may serve as critical determinants influencing sponge-associated microbial diversity and secondary metabolite production [18]. Moreover, the commercial application of sponge-derived bioactive compounds within Kenya is currently underdeveloped, emphasizing the imperative for interdisciplinary collaborations between marine biotechnology researchers and pharmaceutical industries to facilitate the translation of these bioactive molecules into viable therapeutic agents.

This scientific gap impedes the progression of sponge bioactive compounds from laboratory studies to clinical applications, a situation exacerbated by the scarcity of high-throughput screening technologies [19]. Numerous coastal and deep-sea sponge species, particularly those inhabiting African waters, such as those off the coast of Kenya, remain largely underexplored [20]. The World Porifera Database recognizes over 9,700 sponge species globally, with many belonging to the class Demospongiae, which includes species found in Kenya. Additionally, a study identified *Axinella infundibuliformis* from the Kenyan coast, reporting its antimicrobial activity against *Staphylococcus aureus* and *Pseudomonas aeruginosa* [21]. This study aimed to address the scientific gap by identifying and characterizing three marine sponge species, *Biemna fistulosa, Callyspongia diffusa,* and *Haliclona fascigera* from Kenyan coastal waters. Their bioactive potential was assessed through antimicrobial susceptibility testing against *Escherichia coli, Pseudomonas aeruginosa, Staphylococcus aureus,* and *Candida albicans*.

## Materials and methods

### Ethical statement

This research study was officially authorized by the National Commission for Science, Technology and Innovation of Kenya (NACOSTI) under License No: NACOSTI/P/25/4174598. Ethical approval for the study was granted by the Technical University of Mombasa (TUM) under approval No: TUM SERC PhD/006/2025, ensuring that all research activities adhered to institutional ethical standards.

Additionally, the Kenya Marine and Fisheries Research Institute (KMFRI) provided the necessary access permit for field sampling under Reference No: GOK_PC Target C82 39–1/22, facilitating compliance with regulatory requirements for environmental research. Furthermore, the fieldwork did not involve any endangered or protected species, aligning with conservation guidelines and ethical research principles.

### Description of study site

The study sites included Sii Island (4°40'46.1"S, 39°17'01.9"E), Mundini (4°39'10.0"S, 39°21'41.0" E), and Ras Kiromo (4°38'45.5"S, 39°19'23.4"E), located on the South Coastline of Kenya. These are sites characterized by diverse marine ecosystems, with coral reefs and seagrass beds supporting a wide range of aquatic organisms [22].

### Metazoan specimen collection and preparation

A pre-survey for marine sponge sampling was undertaken to determine optimal conditions for clear underwater photography and effective sampling during favorable weather conditions [23]. The collection of metazoan marine sponges took place during the low spring tide, coinciding with the southeast monsoon in September 2022 [24]. The sponge specimens

were collected from depths of 3–5 meters using snorkeling and SCUBA diving techniques [25]. At each site, the sponges were sampled based on the purposive sampling criteria and subsequently rinsed with sterile ocean water to remove debris and epiphytic organisms [26].

A total of 23 marine sponge samples were obtained from six study sites distributed along Kenya's northern and southern coastlines, representing a wide spectrum of taxa differing in abundance and distribution (S1-3 Tables). For the purposes of this study, three sponge species each reflecting a distinct abundance category were chosen for detailed analysis. These samples were collected from different study sites along the Southern coastline (Sii Island, Mundini, and Ras Kiromo) and included *Callyspongia diffusa* (high abundance, found at more than four sites), Biemna fistulosa (moderate abundance, observed at two to three sites), and *Haliclona fascigera* (rare, recorded at a single site).

Specimens were placed in labeled containers filled with sterile ocean water and transported to the Kenya Marine and Fisheries Research Institute (KMFRI) laboratories. Preservation methods were selected based on the intended further analyses: samples were frozen at −20°C for genetic studies and antimicrobial activity assessment, and preserved in 70% ethanol for morphological evaluations [27].

## Morphological characterization of the marine sponges

The sponge morphological identification relied on external morphology, encompassing the color, shape, and surface features [28]. The spicules and spongin fibers within the marine sponge skeleton were digested into small pieces of sponge tissue in bleach and observed under a Primo Star ZEISS image analyzer microscope with a coverslip. The microphotographs were taken using an Axio Cam ERc5s digital camera (Carl Zeiss, Germany) [29]. The marine sponge specimens voucher numbers: BLSi 007, BRMu 004, and BLD 014 used in this study are preserved at the Kenya Marine and Fisheries Research Institute Museum. The marine sponges were coded based on their lifeform color and the study site of collection. The prefixes BL, BR, and BLU denoted black, brown, and blue sponges, respectively. These were followed by Si, Mu, and Ch, representing Sii Island, Mundini, and Ras Kiromo, respectively, along with the specimen number assigned to each sponge from the Kenyan coastline.

The marine sponge species was determined using the World Register of Marine Species (WoRMS) database [30]. To identify the genus and species of marine sponges, the Sponge Identification Reference Book and the Porifera database list (http://www.marinespecies.org/porifera/) were employed [31].

## DNA extraction and PCR amplification of the mitochondrial gene cytochrome c oxidase subunit I (COI)

The salting out method was used for DNA extraction for the sponge specimens [32]. First, 200 μl of the extraction buffer and 5 μl of proteinase K (Thermo Fisher Scientific, Unites States) were added to the tissue. The mixture was incubated at 37°C overnight. Following the incubation, the proteinase K was added, and the mixture was spun down, and 450 μl of 3M sodium chloride (NaCl) was added and vortexed [33]. The sample was then centrifuged at a speed of ≥10,000 g for 15 minutes to collect cell debris. The supernatant, containing DNA (1 ml), was transferred to a sterile 1.5 ml tube and mixed with frozen (−20°C) absolute ethanol to fill the tube [34]. The DNA was allowed to precipitate overnight in a freezer at −20°C, followed by centrifugation at a speed of ≥10,000 g for 20 minutes. The pellet was washed thrice with 700 μl of 70% ethanol and then centrifuged at a speed of ≥10,000 g for another 20 minutes. The ethanol was pipetted out without disturbing the pellet, which was allowed to dry at room temperature [35]. The DNA was then resuspended in 50 μl of deionized water (molecular grade) (Carl Roth, Germany) and its concentration and purity were assessed using a NanoDrop spectrophotometer (Thermo Fisher Scientific, United States).

The degenerate universal barcoding primers dgLCO1490 (GGT CAA CAA ATC ATA AAG AYA TYG G) and dgHCO2198 (TAA ACT TCA GGG TGA CCA AAR AAY CA), targeting the mitochondrial gene cytochrome c oxidase subunit I (COI) were used for amplification [36]. The PCR reaction mixture of 25 μL was prepared by mixing 12.5 μL of 2X PCR Master Mix (DreamTaq Green PCR, Thermo Fisher Scientific, United States), which included Taq polymerase,

dNTPs, and buffer. Additionally, 1.0 µL of both the forward and reverse primers (10 µM) and 2.0 µL of template DNA (20 ng/µL) were added. The reaction volume was then brought to 25.0 µL by adding 8.5 µL of nuclease-free water. For PCR product verification, a 1.5% agarose gel (Sigma-Aldrich, Merck Millipore, Germany) was prepared in 1X TAE buffer, and stain G (SERVA Electrophoresis, Germany) was added [37]. A 5 µL aliquot of the PCR product was mixed with 1 µL of SYBR Green dye (Thermo Fisher Scientific, United States) and loaded into the gel wells. Distilled water was used as the negative control, while the DNA ladder (CSL-MDNA, Cleaver Scientific) was used as a DNA marker. The gel was run at 100V for 30 minutes, and the PCR products were visualized under UV light using a gel documentation system (ATTO, Tokyo, Japan).

The PCR products were treated with an ExoSAP treatment [38]. The purified products were preserved under dry ice conditions and then shipped to Inqaba Biotec, a commercial sequencing service provider in South Africa, for Sanger sequencing.

## DNA sequencing and phylogenetic analysis

Sequencing was conducted using the same primers (dgLCO1490 and dgHCO2198) that were used for PCR amplification. The reaction mixture consisted of the purified DNA template, sequencing primers, BigDye Terminator v3.1 Cycle Sequencing Kit, and buffer [39]. The prepared reaction mixture was subjected to cycle sequencing and then loaded onto an automated DNA sequencer (Applied Biosystems 3500XL Genetic Analyzer). The fluorescence data were interpreted using the FinchTV analysis software to generate the DNA sequence [40].

The raw sequence data from Sanger sequencing were processed using BioEdit software version 7.2. Low-quality bases were identified and removed to maintain data integrity. The sequences were compared against the nucleotide sequence database (GenBank) at the National Center for Biotechnology Information (NCBI) and Barcode of Life Database (BOLD) using the BLASTn algorithm to determine species identity. Additionally, the BOLD platform was utilized for sponge DNA barcoding, focusing on species-level identification through a curated reference library of standardized mitochondrial COI (cytochrome oxidase I) genetic markers [41].

Phylogenetic analysis was conducted using the Unipro UGENE software platform (https://ugene.net/). DNA sequences obtained from GenBank and BOLD databases were together with the newly obtained sequences, were imported in FASTA format and aligned using the MAFFT (Multiple Sequence Alignment using Fast Fourier Transform) algorithm [42]. The Maximum Likelihood method was employed to construct the phylogenetic tree, incorporating bootstrap resampling with 1,000 replications. The resulting phylogenetic tree was visualized using FigTree v1.4.4, with posterior probabilities displayed at the nodes to indicate the support for the branches [43].

## Crude extracts preparation for antimicrobial assays

The sponge samples were freeze-dried and ground into a fine powder using a mechanical grinder [44]. The powdered sponge material was subjected to solvent extraction using methanol, ethyl acetate, and dichloromethane to obtain bioactive compounds with varying polarities [45]. The bioactive compounds recovery involved macerating the sponge powder in the respective solvents for 48 hours at room temperature with occasional agitation [46]. The sponge extracts were then filtered using Whatman No. 1 filter paper (Sigma-Aldrich Company, St. Louis, Germany) and concentrated under reduced pressure using a rotary vacuum evaporator (BIOBASE Company, Jiangsu, China) at temperatures below 40°C to prevent thermal degradation of bioactive compounds [47]. The concentrated crude extracts were weighed and stored in sterile vials at 4°C until further analysis. The sponge extracts were assessed for their antimicrobial efficacy against test microorganisms: *Candida albicans* ATCC 10231, *Pseudomonas aeruginosa* ATCC 27853, *Escherichia coli* ATCC 25922, and *Staphylococcus aureus* ATCC 25923. All microbial strains utilized in this study were procured from the Kenya Medical Research Institute (KEMRI), Kenya.

## Assessment of *In vitro* antimicrobial activity of sponges' crude extracts

The antimicrobial screening of the sponge crude extracts against the test microorganisms was conducted using the Kirby-Bauer disk diffusion method [48]. Sterile Mueller-Hinton agar (MHA) (HiMedia, Mumbai, India) was utilized as the medium for bacterial susceptibility testing [49]. The MHA was prepared following the manufacturer's specifications (38.0 g in 1000 mL of distilled water). For turbidity standardization, the bacterial test organisms were cultured in Mueller-Hinton Broth (MHB) at 37°C for 18 hours with continuous agitation at 150 revolutions per minute (rpm) to ensure uniform bacterial suspension [50]. The MHB (HiMedia, Mumbai, India) was prepared according to the manufacturer's guidelines (21 g in 1000 mL of purified water).

For *C. albicans*, Potato Dextrose Agar (PDA) (HiMedia, Mumbai, India) was used. *C. albicans* broth cultures were incubated at 30°C for 48 hours in Potato Dextrose Broth (PDB), prepared as per the manufacturer's protocol (24 g in 1000 mL of distilled water) for turbidity standardization. Microbial cultures were adjusted to a 0.5 McFarland turbidity standard, representing approximately $1.5 \times 10^8$ CFU/mL for bacterial cells and $1 \times 10^6$ CFU/mL for fungal cells, using sterile normal saline as the diluent. The normal saline was prepared by dissolving 9 g of NaCl in 1000 mL of deionized water [51].

MHA and PDA plates were inoculated with microbial test strains employing a sterile swab technique. Sponge crude extracts were solubilized in dimethyl sulfoxide (DMSO) to achieve working concentrations of 10 µL of 10 mg/mL sponge extracts in DMSO. Sterile paper discs (Oxoid, Thermo Scientific, United Kingdom) measuring 6 mm in diameter, were impregnated with 20 µL of sponge extracts and positioned on the inoculated agar plates [52]. Streptomycin (200 µg/ml) served as the positive control for bacterial strains, whereas fluconazole (10 µg/mL) was employed as the positive control for *C. albicans*, with DMSO functioning as the negative control [53]. Following incubation, plates were maintained at 37°C for bacterial strains and at 30°C for fungal strains, with measurements of inhibition zone diameters (IZD) conducted in millimeters using a digital caliper to evaluate antimicrobial efficacy [54].

## Determination of the minimum inhibitory concentration (MIC)

The broth microdilution method was employed to evaluate the antimicrobial efficacy of the marine sponge extracts against the selected pathogenic microbial strains [55]. This method involved the preparation of a two-fold serial dilution of the marine sponge extracts in nutrient broth for antibacterial screening and in PDB for antifungal assessment [56]. The test tubes were systematically labelled (Tube 2, Tube 3, Tube 4, Tube 5, Positive Control (+ve), and Negative Control (-ve). A stock solution of the sponge extract (Tube A) was initially prepared for subsequent dilutions. A volume of 2 mL of sterile broth was dispensed into each labelled test tube, beginning with the negative control. For the serial dilution, 2 mL of the sponge extract from Tube A was transferred into Tube 2 and mixed thoroughly. Subsequently, the process was repeated sequentially for Tubes 3–5. The concentration gradient of the stock solution in the nutrient broth resulted in final concentrations of 0.625 mg mL$^{-1}$, 1.25 mg mL$^{-1}$, 2.5 mg mL$^{-1}$, 5 mg mL$^{-1}$, and 10 mg mL$^{-1}$ of the sponge extracts [57].

Following the preparation of the serial dilutions, 0.3 mL of the microbial suspension (*E. coli*, *S. aureus*, *P. aeruginosa*, and *C. albicans*), was introduced into all tubes except the negative control [58]. The test tubes were then incubated under aerobic conditions at 37°C for 18 hours for bacterial strains and at 30°C for 48 hours for *C. albicans* to allow for microbial growth [59]. After the incubation period, bacterial growth was assessed by measuring the optical density (OD) at 600 nm using a spectrophotometer (Shimadzu, Japan) [60]. The minimum inhibitory concentration (MIC) was determined as the lowest concentration of the sponge extract that visibly inhibited bacterial and fungal growth [61].

## Evaluation of the minimum bactericidal concentration (MBC) and minimum fungicidal concentration (MFC)

The minimum bactericidal (MBC) and fungicidal (MFC) concentrations were determined for sponge extracts that exhibited low MICs against the tested microorganisms. The tests were done through the subculturing of wells from the MIC assays onto fresh agar plates [62]. Specifically, the bacterial cultures were incubated for 18 hours at 37 °C [63], while *C. albicans*

cultures were incubated at 30 °C for 48 hours [64]. Following incubation, the MHA plates were examined for the presence of surviving organisms. If the MBC of the sponge extracts was less than or equal to the MBC of the standard drug, the extract's activity was classified as either bactericidal or fungicidal [65].

## Gas chromatography-mass spectrometry (GC-MS) of sponge crude extracts

The dried sponge extracts were reconstituted in analytical-grade n-hexane of high purity to attain a final concentration of 1 mg mL$^{-1}$ [66]. Subsequently, the reconstituted samples were subjected to filtration through a 0.22 μm membrane filter to eliminate any particulate contaminants. A 1 μL aliquot of the prepared extract was injected into the GC-MS system for analytical assessment [67]. The GC-MS system was equipped with an HP-5MS capillary column (5% phenyl methylpoly-siloxane, 30 m × 0.25 mm i.d. × 0.25 μm film thickness) [68]. High-purity helium served as the carrier gas at a constant flow rate of 1 mL/min, and the analysis was performed in splitless mode to enhance sensitivity. The injector temperature was maintained at 250°C, while the oven temperature program commenced at an initial temperature of 50°C, held for 2 minutes, followed by a ramp rate of 10°C/min up to a final temperature of 300°C, where it was sustained for 5 minutes [69].

The mass spectrometer was operated in Electron Ionization (EI) mode at an electron energy of 70 eV, with ion source temperature of 230°C and a quadrupole temperature of 150°C. The mass scan range was configured from 50 to 600 m/z, with a solvent delay period of 3 minutes. The mass spectra of the compounds detected in this study were systematically compared against established reference spectra derived from the National Institute of Standards and Technology (NIST) libraries [70]. An analysis of retention times and molecular ion peaks was conducted to authenticate the identities of the compounds. The relative abundance of bioactive compounds was quantified through peak area integration, enabling a precise determination of their prevalence. Identified compounds were categorized according to their chemical classes, including alkaloids, terpenoids, steroids, and fatty acids. The results were compared with previously reported marine-derived bioactive compounds to assess their novelty and potential pharmaceutical applications [14].

## Data analysis

A two-way analysis of variance (ANOVA) was performed to determine the presence of statistically significant differences in the diameters of inhibition zones across a sample size of twenty-three marine sponge extracts. The data were recorded in triplicate and presented as mean ± standard deviation (SD) for the inhibition zones. Statistical tests of means sharing identical superscripts (*) within the row were considered not significantly different from one another, as determined through Fisher's Least Significant Difference (LSD) test at a 95% confidence level, following a post hoc analysis conducted using Minitab software version 21.4.1. Differences were classified as statistically significant at P < 0.05 (α = 0.05).

Mass spectra generated from the GC-MS analysis were compared against entries in the NIST 14 Mass Spectral Library, with only those exhibiting match quality scores above 75% considered valid for interpretation. Where applicable, these initial identifications were corroborated using published literature and retention index references.

## Results

### Morphology and taxonomy of marine sponges

These sponge samples were collected from Sii Island, Mundini, and Ras Kiromo study sites and taxonomically identified as *Biemna fistulosa, Callyspongia diffusa,* and *Haliclona fascigera*, respectively. Upon removal from their natural habitat, the sponges exhibited significant morphological changes, including pronounced shrinkage and increased fragility. Additionally, a notable decline in their vibrant coloration was observed (Fig 1).

*Biemna fistulosa* sponge exhibited a robust, spongy and fibrous texture, with a growth pattern that encrusted and branched upwards (Fig 2). Initially, it presented a white-grey hue underwater, which upon exposure to air, turned black. The species, *B. fistulosa*, typically inhabited sandy shores and mangrove lagoons. Its structure was characterized by densely

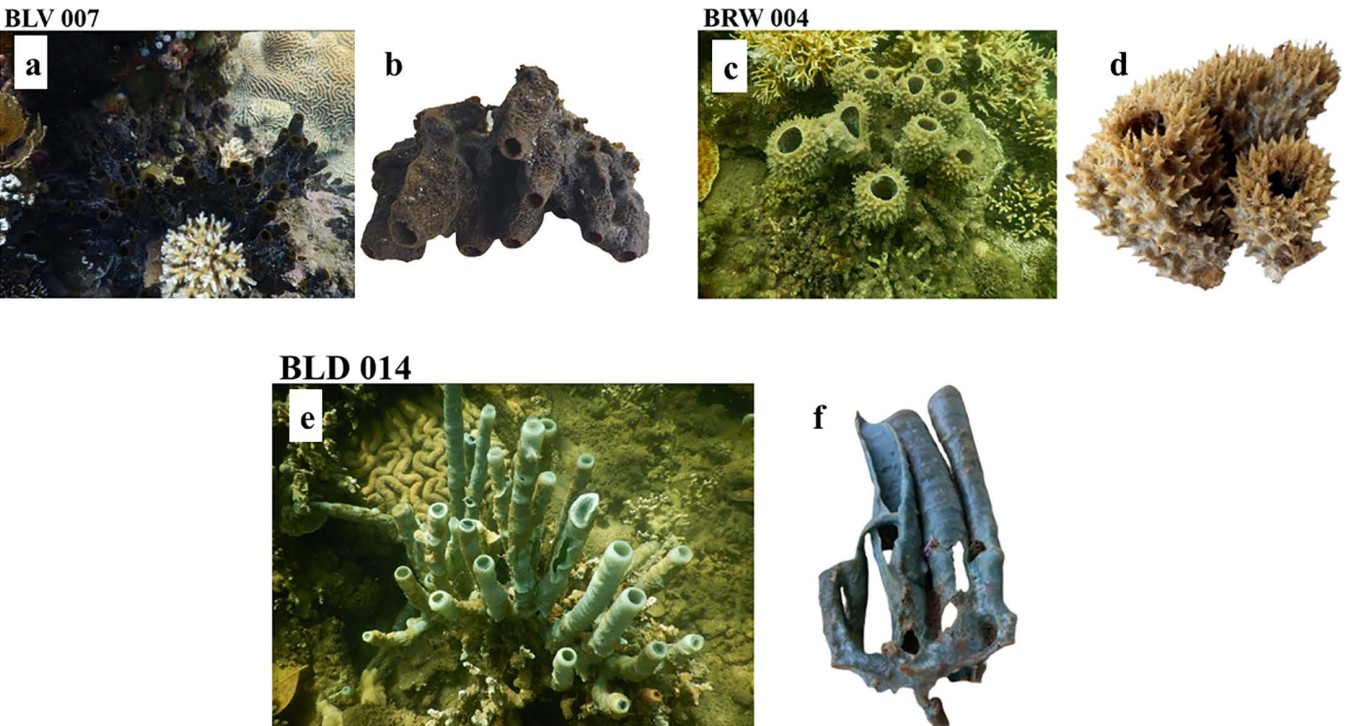

**Fig 1. Photographic documentation of marine sponges from the Kenyan coastline, presented as follows: (a)** *Biemna fistulosa* (Voucher specimen (BLSi 007) (in-situ) **(b)** *B. fistulosa* (detached) **(c)** *Callyspongia diffusa* (Voucher specimen BRMu 004) (in-situ) **(d)** *C. diffusa* (detached) **(e)** *Haliclona fascigera* (Voucher specimen BLUCh 014) (in-situ) **(f)** *Haliclona fascigera* (detached) (*Source: Author*).

packed diactinal curved in styles (685.0-970.9-1235.7 x 7.9-19.0-32.8 µm). These spicules formed an irregular network, creating polygonal patterns. Additionally, microscleres such as C-shaped sigmas (30.4-41.0-50.3 µm), were also observed.

*Callyspongia diffusa* also called the diffuse rope sponge exhibited a main skeleton composed of an isodictyl reticulation of collagenous spongin fibers, with diactinal spicules forming triangular meshes in the skeleton (Fig 3). The *C. diffusa* megascleres were represented by anchorates, acanthostyles (209.0-224.5-267.0 x 5.7-7.9-11.9 µm) and strongyles (163.9-178.2-179.3 x 4.9-7.4-10.7 µm), while the microscleres took the form of sterrasters. Additionally, the sponge was characterized by numerous raphides (149.3-168.6-279.8 µm).

The sponge *Haliclona fascigera* possessed a choanosomal skeleton, characterized by an isodictyal network of spicules intertwined with nodal spongin fibers, creating polygonal patterns (Fig 4). Its primary structural megascleres were exclusively oxeas (428-497.9-634.1 x 8.3-13.1-20.0 µm) and were embedded within spongin fibers. A significant number of raphides (114.2-163.0-179.3 µm), microstrongyles (128.0-148.0-178.4 µm) and synapta plates were observed. *H. fascigera* typically inhabited the shallow, sandy bottoms of lagoons and the vicinities of coral reefs.

## Taxonomic affiliation of the marine sponges

The genomic DNA extracted from marine sponge samples had NanoDrop quantifications ranging between 84.9 and 245.3 µg/mL. PCR amplification of the cytochrome c oxidase subunit I (COI) gene yielded amplicons within the expected size range of approximately 650–700 base pairs (bp). All sponge specimens were classified within the phylum Porifera and identified as belonging to the genera *Callyspongia, Haliclona*, and *Biemna* based on percentage sequence similarities. A comparative analysis of the newly obtained mitochondrial Cytochrome Oxidase Subunit 1 (CO1) sequences (PQ329108,

**Fig 2. Morphological and skeletal characterization of *Biemna fistulosa* (Voucher specimen BLSi 007). A. Marine poriferan BLSi 007,** *Biemna fistulosa;* **B. sponge skeleton:** *(1sa)*: perpendicular section; and *(1sb)*: a tangential section (40x magnification); **C. sponge spicules:** *(1sc)*: synapta plates; *(1sd)* and *(1se)*: megascleres acanthostyles; *(1sf)*: pentactines megascleres with digits at the tentacles; *(1sg)*: curved oxeas; *(1sh)*: curved styles; *(1si)* and *(1sk)*: stauractines megascleres with digits at the tentacles; *(1sj)*: pentactines megascleres with digits at the tentacles; *(1sl)*: styles; (1 sm): tabulated strongyles; *(1sn)*: raphides; *(1so)*: Microbiota (*Coscinodiscus radiatus)*; *(1sp)* and *(1sr)*: sterrasters; *(1sq)*: microstrongyles; *(1ss)*: strongyles; and *(1st)*: dendroclones (unique to extinct sponge); **D. spongin fibers:** *(1fa)*: simple elongated spongin fibers; *(1fb)*: simple irregular spongin fibers; *(1fc)*: Spongin fiber forming a fiber network on one end; *(1fd)*: Spongin fiber with an irregular shape; *(1fe)*: spongin fiber with a curved structure *(Source: Author)*.

PQ997929, and PQ997931), conducted using BLASTn search against the GenBank database, showed sequence similarities of ≥99% with existing entries in the nucleotide sequence repository (Table 1). Each sponge sample formed a distinct sub-cluster (representing a specific genus), with a bootstrap support value of 100% (Table 1 and Fig 5). The marine sponge specimens BRMu 004 (PQ329108) from Mundini and BLUCh 014 (PQ997929) from Ras Kiromo were closely related to *Callyspongia diffusa* and *Haliclona fascigera*, respectively. These specimens had sequence identities of 100% and 99%, respectively (Table 1), and formed distinct subclusters, each supported by a bootstrap value of 100%, as depicted in the phylogenetic tree (Fig 5). The sponge specimen BLSi 007 (PQ997931) from Sii Island was closely related to the known sponge species *Biemna fistulosa*, with a sequence identity of 98%, and formed a subcluster with a bootstrap support value of 100% (Table 1 and Fig 5).

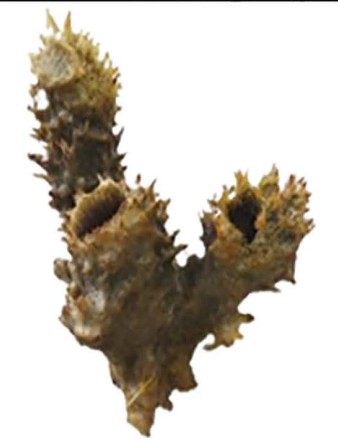
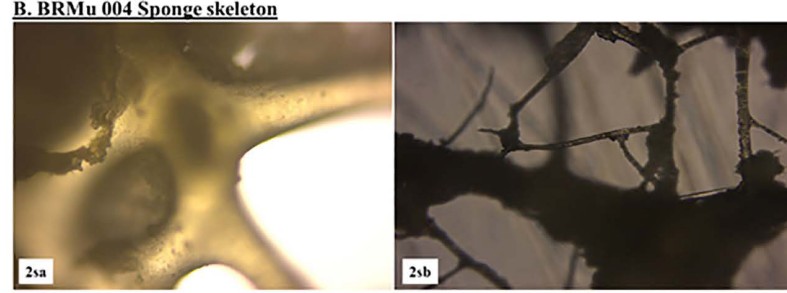
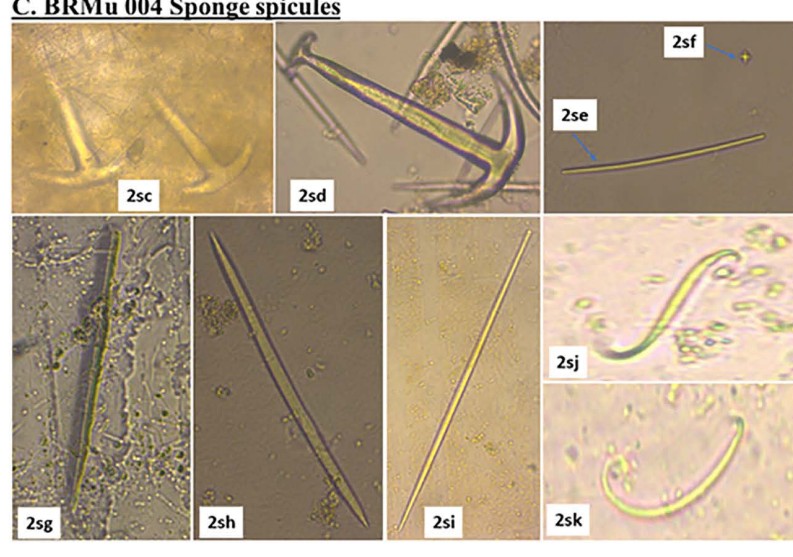
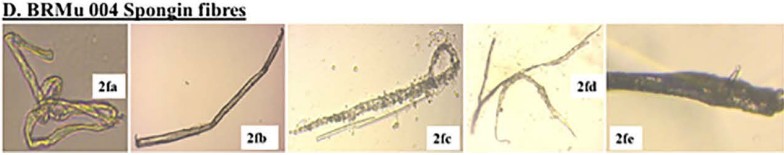

**Fig 3. Skeletal architecture and spicule morphology of *Callyspongia diffusa* (Voucher specimen BRMu 004). A. Marine poriferan BRMu 004,** *Callyspongia diffusa*; **B. sponge skeleton:** *(2sa):* perpendicular section; and *(2sb):* a tangential section (40x magnification); **C. sponge spicules:** *(2sc)* and *(2sd):* anchorates; *(2se):* strongyles; *(2sf):* sterrasters; *(2sg):* acanthostyles; *(2sh):* curved oxeas; *(2si):* styles; *(2sj):* S sigmas; and *(2sk):* C sigmas; **D. spongin fibers:** *(2fa):* twisted thick spongin fibers; *(2fb):* Spongin fibers with bent thickened cell walls; *(2fc):* spongin fibers with a complete bent (*Microcoleus vaginatus* attaching on the surface); *(2fd):* spongin fibers with an anastomosing system; and *(2fe):* spongin fibers with hard collagen material (spicules protruding) *(Source: Author).*

### *In vitro* antibiotic and antifungal activity of the marine sponge crude extracts

The marine sponge extracts were evaluated for their effectiveness against human pathogenic strains (*Candida albicans,* ATCC 10231, gram-negative bacteria *Pseudomonas aeruginosa* ATCC 25923 and *Escherichia coli* ATCC 25922 and the gram-positive bacterium *Staphylococcus aureus* ATCC 27853 (Tables 2–5). All the three crude organic extracts from the marine sponges (*B. fistulosa, C. diffusa* and *H. fascigera*) showed significant antimicrobial activity against at least one of the four tested microorganisms, exceeding the efficacy of the positive control (P<0.05) (Tables 2–5).

The methanolic extracts from the sponges *B. fistulosa* (28.00±3.5 mm) and *H. fascigera* (28.33±3.8 mm) exhibited a broad spectrum of antibacterial activity against *E. coli* that was statistically higher than the positive control (27.67±0.9 mm)

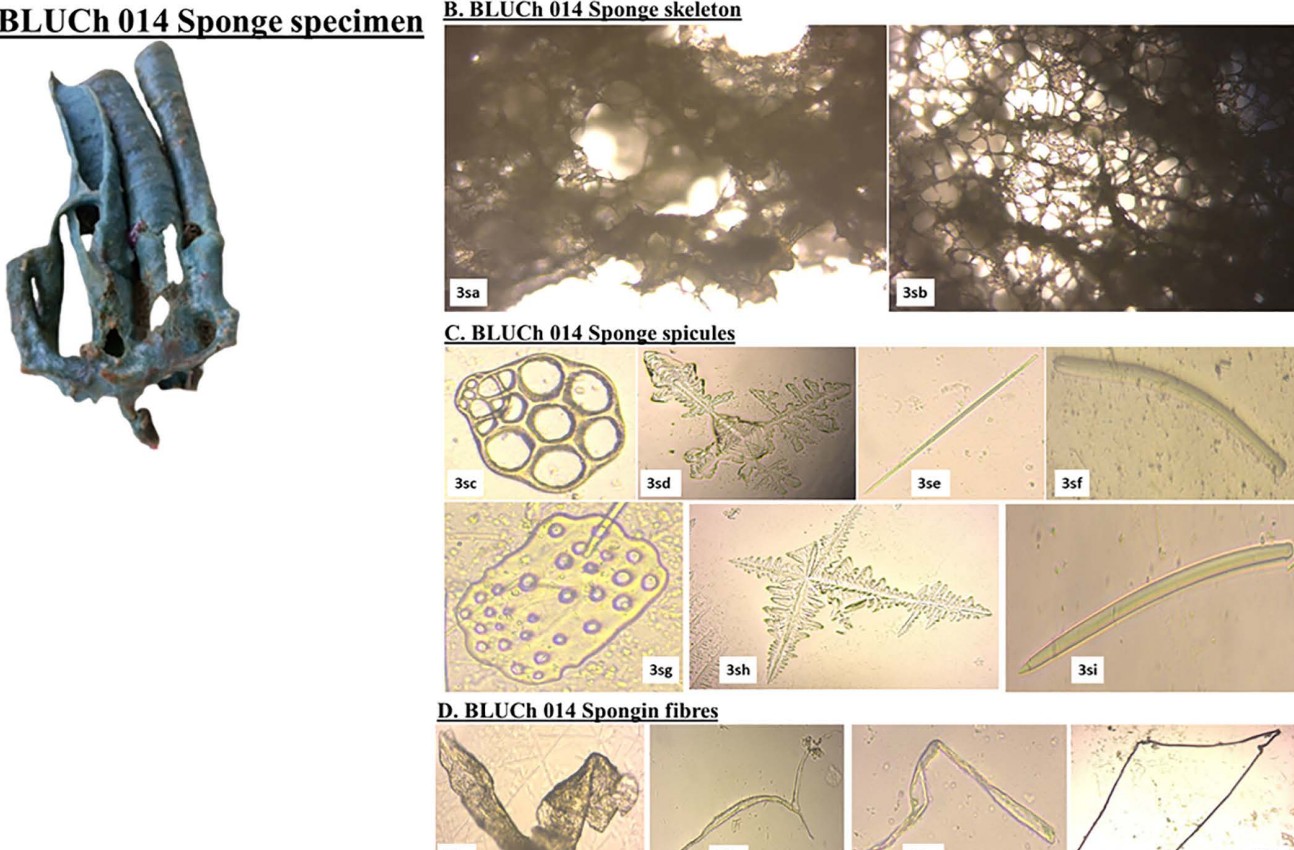

**Fig 4. Skeletal composition and spicule diversity of *Haliclona fascigera* (Voucher specimen BLUCh 014). A.** Marine poriferan BLUCh 014, *Haliclona fascigera*; **B. sponge skeleton:** *(3sa):* perpendicular section; and *(3sb):* a tangential section (40x magnification); **C. sponge spicules:** *(3sc):* synapta plates; *(3sd)* and *(3sh):* stauractines megascleres with digits at the tentacles; *(3se);* styles; *(3sf):* tuberculated curved strongyles; *(3sg):* large plates of calcareous deposits; *(3si):* styles; **D. spongin fibers:** *(3fa):* spongin fibers with a thick flat structure; *(3fb):* branched spongin fibers; *(3fc):* twisted spongin fibers with an open transparent lumen; *(3fd):* spongin fibers with thickened cell walls and a smooth transparent lumen *(Source: Author).*

**Table 1. Taxonomic affiliation of marine metazoan sponges with their closest phylogenetic relatives.**

| Sample ID | Accession No. | Location | Closest taxonomic affiliation | Isolation Source | Country | % ID |
|---|---|---|---|---|---|---|
| BLSi 007 | PQ997931 | Sii Island (4°40'46.1"S, 39°17'01.9"E) | *Biemna fistulosa* (AM076982.1) | Coral reefs [79] | United States | 98 |
| BRMu 004 | PQ329108 | Mundini (4°39'10.0"S, 39°21'41.0"E) | *Callyspongia diffusa* (KX454494.1) | Indo-Pacific region [73] | India | 100 |
| BLUCh 014 | PQ997929 | Ras Kiromo (4°38'45.5"S, 39°19'23.4"E) | *Haliclona fascigera* (OQ322782.1) | Island [15] | Indonesia | 99 |

(Table 2). The inhibitory activity of ethyl acetate extracts *C. diffusa)* against *P. aeruginosa* was observed to be statistically higher (29.33 ± 2.4 mm) compared to that of the positive control (26.67 ± 0.7 mm) (Table 3).

*B. fistulosa* exhibited the strongest antibacterial activity against *S. aureus* in methanolic extracts (25.33 ± 0.9 mm), whereas *H. fascigera* showed the highest potency in ethyl acetate extracts (24.67 ± 1.2 mm). Despite their notable inhibitory potential, both sponges were less effective than the standard antibiotic, streptomycin (31.11 ± 0.2 mm) (Table 4). Methanolic extracts of *B. fistulosa* exhibited a higher antifungal activity (15.67 ± 1.2 mm) against *C. albicans* compared to the

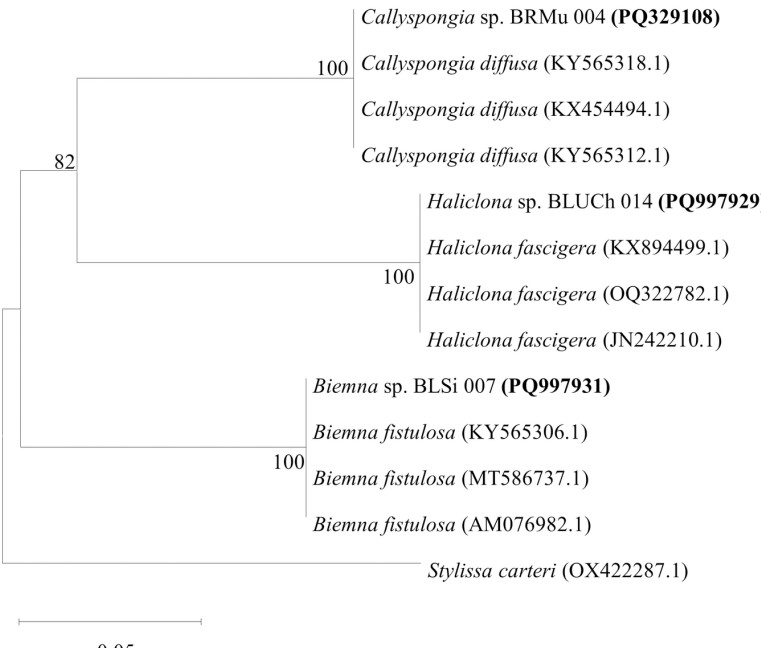

*Callyspongia* sp. BRMu 004 **(PQ329108)**

*Callyspongia diffusa* (KY565318.1)

*Callyspongia diffusa* (KX454494.1)

*Callyspongia diffusa* (KY565312.1)

*Haliclona* sp. BLUCh 014 **(PQ997929)**

*Haliclona fascigera* (KX894499.1)

*Haliclona fascigera* (OQ322782.1)

*Haliclona fascigera* (JN242210.1)

*Biemna* sp. BLSi 007 **(PQ997931)**

*Biemna fistulosa* (KY565306.1)

*Biemna fistulosa* (MT586737.1)

*Biemna fistulosa* (AM076982.1)

*Stylissa carteri* (OX422287.1)

0.05

**Fig 5. Phylogenetic Relationships of Metazoan CO1 Sequences with Closely Related Sponge Species.** The phylogenetic tree was rooted using *Stylissa carteri* (OX422287.1). Bootstrap values exceeding 50%, derived from 1000 replications, are indicated at the branch nodes. The scale bar represents 0.05 substitutions per nucleotide.

**Table 2. Antimicrobial activities of organic crude extracts from the selected marine sponges against *Escherichia coli*.**

| Sample ID | Sponge Species | Marine sponge extracts (mm) | | |
|---|---|---|---|---|
| | | Dichloromethane | Methanolic | Ethyl Acetate |
| BLSi 007 | *Biemna fistulosa* | 9.00 ± 1.2$^g$ | 28.00 ± 3.5$^{abcd}$ | 8.33 ± 0.3 cd |
| BRMu 004 | *Callyspongia diffusa* | 12.33 ± 0.9$^g$ | 0.00 ± 0.0$^d$ | 13.00 ± 2.1$^d$ |
| BLUCh 014 | *Haliclona fascigera* | 8.00 ± 0.6$^g$ | 28.33 ± 3.8$^{abcd}$ | 8.67 ± 1.2 cd |
| Positive control | | 31.67 ± 0.7$^{ab}$ | 27.67 ± 0.9$^a$ | 27.67 ± 0.9$^a$ |
| Negative | | 0.00 ± 0.0 $^g$ | 0.00 ± 0.0$^d$ | 0.00 ± 0.0$^d$ |

**Table 3. Antimicrobial activities of organic crude extracts from the selected marine sponges against *Pseudomonas aeruginosa*.**

| Sample ID | Sponge Species | Marine sponge extracts (mm) | | |
|---|---|---|---|---|
| | | Dichloromethane | Methanolic | Ethyl Acetate |
| BLSi 007 | *Biemna fistulosa* | 15.33 ± 1.5$^{cdefg}$ | 19.67 ± 1.2$^{abc}$ | 0.00 ± 0.0$^e$ |
| BRMu 004 | *Callyspongia diffusa* | 16.00 ± 1.5$^g$ | 0.00 ± 0.0$^c$ | 29.33 ± 2.4$^{cde}$ |
| BLUCh 014 | *Haliclona fascigera* | 7.67 ± 0.7$^{efg}$ | 15.67 ± 0.7$^{ab}$ | 8.33 ± 0.3$^e$ |
| Positive control | | 31.22 ± 0.6$^{ab}$ | 26.67 ± 0.7$^{ab}$ | 26.67 ± 0.7$^{abcd}$ |
| Negative | | 0.00 ± 0.0$^g$ | 0.00 ± 0.0$^c$ | 0.00 ± 0.0$^e$ |

**Table 4. Antimicrobial activities of organic crude extracts from the selected marine sponges against *Staphylococcus aureus*.**

| Sample ID | Sponge Species | Marine sponge extracts (mm) | | |
| --- | --- | --- | --- | --- |
| | | Dichloromethane | Methanolic | Ethyl Acetate |
| BLSi 007 | *Biemna fistulosa* | 13.30 ± 1.5[de] | 25.33 ± 0.9[abc] | 0.00 ± 0.0[d] |
| BRMu 004 | *Callyspongia diffusa* | 0.00 ± 0.0[e] | 0.00 ± 0.0[d] | 9.33 ± 0.9[d] |
| BLUCh 014 | *Haliclona fascigera* | 9.33 ± 0.9[de] | 24.67 ± 1.2[abc] | 24.67 ± 1.2 cd |
| Positive control | | 31.11 ± 0.2[ab] | 31.11 ± 0.2[abc] | 31.11 ± 0.2[abc] |
| Negative | | 0.00 ± 0.0[e] | 0.00 ± 0.0[d] | 0.00 ± 0.0[d] |

methanolic extract of *H. fascigera* (10.33 ± 0.9 mm) and *C. diffusa* (9.67 ± 2.2 mm), however, its activity did not exceed that of the positive control (25.67 ± 1.2 mm) (Table 5). None of the dichloromethane and ethyl acetate extracts of the sponges exhibited antifungal activity against *C. albicans* (Table 5).

## Evaluation of the sponge extracts for MIC, MBC, and MFC

The Minimum Inhibitory Concentrations (MICs) of the sponge extracts were evaluated, demonstrating a range of values from 0.625 mg mL$^{-1}$ to 10 mg mL$^{-1}$. Streptomycin demonstrated the lowest MIC compared to the sponge extracts, with values of 1.36 mg/ml against *E. coli* and 1.94 mg/ml against *P. aeruginosa* (Table 6). The lowest MIC value (0.53 ± 0.01 mg mL$^{-1}$) of methanolic extract of isolate *H. fascigera* was observed against *E. coli* compared to the standard drug/control (1.36 ± 0.00 mg mL$^{-1}$) (Table 6). The MBC for the standard streptomycin drug was established at 2.5 mg mL$^{-1}$.

Within each group, means with identical superscript letters indicate no significant difference at a 95% confidence level (α = 0.05), as determined by Fisher's Least Significant Difference (LSD) test.

The organic crude extract obtained from *H. fascigera* demonstrated the most potent bactericidal activity, with a minimum bactericidal concentration (MBC) of 1.25 mg mL$^{-1}$, exceeding the efficacy of the standard drug streptomycin as well as the extracts derived from *C. diffusa* and *B. fistulosa*. Specifically, the MBC values of the sponge extracts were recorded as 2.5 mg mL$^{-1}$ for *C. diffusa* and 5 mg mL$^{-1}$ for *B. fistulosa*. In comparison, the MBC of the streptomycin reference drug was established at a 2.5 mg mL$^{-1}$ concentration. The methanolic extract of *H. fascigera* demonstrated the most potent fungicidal activity, with a minimum fungicidal concentration (MFC) of 2.5 mg mL$^{-1}$. Notably, the extracts from both sponges *C. diffusa* and *B. fistulosa* exhibited an MFC of 10 mg mL$^{-1}$. The fluconazole reference drug showed an MFC comparable to that of *H. fascigera*, with a value of 2.5 mg mL$^{-1}$ concentration.

## GC-MS spectral analysis of the crude extract of the marine sponge extracts

The GC-MS analysis of the extracts from *B. fistulosa, C. diffusa,* and *H. fascigera* generated a spectral profile and chemical structures of the detected compounds (Table 7). The peak numbers in the chromatograms correspond to the identified

**Table 5. Antimicrobial activities of organic crude extracts from the selected marine sponges against *Candida albicans*.**

| Sample ID | Sponge Species | Marine sponge extracts (mm) | | |
| --- | --- | --- | --- | --- |
| | | Dichloromethane | Methanolic | Ethyl Acetate |
| BLSi 007 | *Biemna fistulosa* | 0.00 ± 0.0[c] | 15.67 ± 1.2[abc] | 0.00 ± 0.0[c] |
| BRMu 004 | *Callyspongia diffusa* | 0.00 ± 0.0[c] | 9.67 ± 2.2 cd | 0.00 ± 0.0[c] |
| BLUCh 014 | *Haliclona fascigera* | 0.00 ± 0.0[c] | 10.33 ± 0.9[abc] | 0.00 ± 0.0[c] |
| Positive control | | 29.33 ± 1.5[ab] | 25.67 ± 1.2[ab] | 25.67 ± 1.2[ab] |
| Negative | | 0.00 ± 0.0[c] | 0.00 ± 0.0[d] | 0.00 ± 0.0[c] |

**Table 6. Minimum inhibitory concentrations (MIC) of dichloromethane, methanolic, and ethyl acetate organic crude extracts of the selected marine sponges against the tested human pathogens.**

| | | | MIC (mg/ml) | |
|---|---|---|---|---|
| **Sample ID** | **Marine sponge** | **Extract** | *Escherichia coli* | *Pseudomonas aeruginosa* |
| BLSi 007 | *Biemna fistulosa* | Methanolic | 2.46±0.01[a] | – |
| BRMu 004 | *Callyspongia diffusa* | Ethyl acetate | – | 2.63±0.01[a] |
| BLD 014 | *Haliclona fascigera* | | 0.53±0.01[a] | – |
| Positive Control | | | 1.36±0.00[a] | 1.94±0.00[a] |
| Negative Control | | | 0.00±0.00 | 0.00±0.00 |

compounds (Figs 6–8 and Table 8). The GC-MS chromatogram data identified a total of 114 compounds across the three sponge extracts (BLSi 007, BRMu 004, BLUCh 014). These compounds belong to 39 distinct chemical classes (Table 7).

The methanolic extract of the marine sponge *B. fistulosa* (BLSi 007) revealed a total of 47 chemical compounds from the GC-MS analysis (Fig 6). These compounds were grouped into alkane derivatives (2.1%), amino acid derivatives (6.4%), cyclic alkanes (2.1%), cyclic amines (2.1%), cyclic dipeptides (2.1%), cyclic esters (2.1%), diketones (2.1%), diazabicyclo compounds (2.1%), ergot alkaloids (4.3%), ester derivatives (6.3%), esterified fatty acids (10.6%), esterified forms of ascorbic acids (vitamin C) (2.1%), fatty acids (2.1%), heterocyclic amines (10.6%), heterocyclic compounds (10.6%), hydrazones (2.1%), imidazoles (2.1%), ketones (4.3%), long-chain saturated fatty acids (4.3%), monounsaturated fatty acids (10.6%), organic nitrogenous compounds (2.1%), pyrazole derivatives (2.1%), triazole derivatives (2.1%), and unsaturated fatty acids (2.1%).

The GC-MS chromatogram data identified a total of 62 chemical compounds from the ethyl acetate extract of the sponge *C. diffusa* (BRMu 004) (Fig 7). The detected compounds were classified into alkenes (1.6%), amino acid derivatives (9.5%), amines (4.8%), aromatic hydrocarbons (4.8%), carboxylic acid anhydrides (1.6%), chiral amino alcohols (1.6%), chiral organic compounds (1.6%), esters (12.7%), ergot alkaloids (1.6%), fatty alcohols (1.6%), heterocyclic organic compounds (27.0%), ketals (1.6%), organic esters (15.9%), phthalic acid esters (7.9%), polycyclic hydrocarbons (1.6%), quinolinedione derivatives (1.6%), triglycerides (1.6%), and α-ketoglutaric acids (1.6%).

The GC-MS analysis of the methanolic extract from the marine sponge *H. fascigera* (BLUCh 014) identified 37 chemical compounds (Fig 8). The chemical compound were categorized as acetylated amine alcohols (1.4%), amide compounds (2.7%), amide ester derivatives (13.5%), alkynes (2.7%), amino acid derivatives (2.7%), bicyclic chemical compounds (2.7%), cyclic alcohols (2.7%), cyclic alkanes (5.4%), cyclic amines (2.7%), epoxides (2.7%), esters (18.9%), heterocyclic amines (5.4%), heterocyclic boron-containing compounds (2.7%), heterocyclic compounds (10.8%), medium-chain fatty acids (2.7%), protected sugar derivatives (2.7%), pyrazole derivatives (2.7%), spiro compounds (2.7%), sugar alcohols (5.4%), synthetic compounds (2.7%), triazole derivatives (2.7%), and unsaturated carboxylic acids (2.7%).

In this study, GC-MS chromatogram analysis of the methanolic extract from *Biemna fistulosa* (BLSi 007) identified four potent bioactive compounds: (2S,6R)-2,6-dibutyl-4-methylpiperidine, pyrrolo [1,2-a] pyrazine-1,4-dione, hexahydro-3-(phenylmethyl), 1-(2-ethyl-1,2,4-triazol-3-yl) ethanamine, and 3,6-diisopropylpiperazin-2,5-dione (Fig 6 and Table 8).

Similarly, GC-MS analysis of the ethyl acetate extract from *Callyspongia diffusa* (BRMu 004) revealed six secondary bioactive compounds, including 7-n-pentadecylaminomethyl-6-hydroxy-5,8-quinolinedione, eicosanoic acid butyl ester, pyrrolo [1,2-a] pyrazine-1,4-dione, hexahydro-3-(phenylmethyl), 4H-pyran-4-one, 2,2'-isopropylidenebis [3-methoxy-6-methyl], (2S,6R)-2,6-dibutyl-4-methylpiperidine, and phthalic acid, 2-ethylhexyl tetradecyl ester (Fig 7 and Table 8).

The methanolic extract from *Haliclona fascigera* (BLUCh 014) exhibited five bioactive compounds: xylitol, 1-(2-ethyl-1,2,4-triazol-3-yl) ethanamine, pyrrolo [1,2-a] pyrazine-1,4-dione, hexahydro-3-(phenylmethyl), (2S,6R)-2,6-dibutyl-4-methylpiperidine, and n-propyl 9-tetradecenoate (Fig 8 and Table 7). Notably, the bioactive compounds (2S,6R)-2,6-dibutyl-4-methylpiperidine and pyrrolo [1,2-a] pyrazine-1,4-dione, hexahydro-3-(phenylmethyl) were consistently present in all three sponge extracts.

**Table 7. Classification of marine sponge compounds identified via GC-MS in organic extracts from *Biemna fistulosa*, *Callyspongia diffusa*, and *Haliclona fascigera* collected from Kenyan waters.**

| No. | Compound chemical class | Marine sponge compounds |
|---|---|---|
| 1. | α-Ketoglutaric acids | 2-Oxopentanedioic acid |
| 2. | Alkenes | 1-Dodecene |
| 3. | Alkynes | 2-Octyne, 1,1-diethoxy |
| 4. | Amide ester derivatives | L-Proline, N-valeryl-, pentadecyl ester, |
| 5. | Amines | Tris(dimethylamino)methane, 2-Propanamine, 2-methyl-N-(phenylmethylene)-, N-oxide, Propanamide, 2,2-dimethyl-N-(2,6-dimethylphenyl), 1,4-Pentanediamine, N1, N1-diethyl |
| 6. | Amino acid derivatives | l-Proline, N-allyloxycarbonyl-, propyl ester, L-Proline, N-valeryl-, hexadecyl ester, L-Proline, N-valeryl-, tetradecyl ester, l-Leucine, N-cyclopropylcarbonyl-, pentadecyl ester, l-Leucine, N-cyclopropylcarbonyl-, undecyl ester, 3-Pyrrolidin-2-yl-propionic acid |
| 7. | Aromatic hydrocarbons | Azulene, Naphthalene, 1H-Indene, 1-methylene, Benzene, 1,3-diethyl-5-methyl, Cyclopropanemethanol, 1-phenyl-, 2-Isopropylbenzaldehyde, Benzene, 2,4-diethyl-1-methyl-, Benzene, 1,4-diethyl-2-methyl-, Benzene, 1-ethyl-3-(1-methylethyl)-, Benzene, 1,3-dimethyl-5-(1-methylethyl)-, Benzene, 2,4-diethyl-1-methyl-, 3,4-Dimethylcumene |
| 8. | Bicyclic chemical compounds | Isosorbide, |
| 9. | Carboxylic acid anhydrides | 2-Dodecen-1-yl (-) succinic anhydride, 3-Heptenoic acid |
| 10. | Chiral amino alcohols | 2-Pyrrolidinemethanol, 2-methyl-, (S) |
| 11. | Cyclic alcohols | Bicyclo [3.1.1] heptan-3-ol, 2,6,6-trimethyl-, [1R-(1. alpha.,2. beta.,3. alpha.,5. alpha.)] |
| 12. | Cyclic alkanes | Cyclobutane, 2-hexyl-1,1,4-trimethyl-, cis, Cyclopropane, 1-methyl-1-(1-methylethyl)-2-nonyl |
| 13. | Cyclic amines | diketone |
| 14. | Cyclic dipeptides | 3,6-Diisopropylpiperazin-2,5-dione |
| 15. | Diazabicyclo compounds | 3-Methyl-1,4-diazabicyclo [4.3.0] nonan-2,5-dione, N-acetyl |
| 16. | Epoxides | Oxirane, 2-methyl-2-(1-methylpropyl) |
| 17. | Ergot alkaloids | Ergotamine, Dihydroergotamine |
| 18. | Esterified fatty acids | Octadecanoic acid, 2-(2-hydroxyethoxy) ethyl ester, Dodecanoic acid, 2-octyl, trans-9-Octadecenoic acid, pentyl ester |
| 19. | Esterified form of ascorbic acid (vitamin C) | l-(+)-Ascorbic acid 2,6-dihexadecanoate |
| 20. | Fatty acids | Decanoic acid, 10-(2-hexylcyclopropyl), Eicosanoic acid, Pentadecanoic acid, Octanoic acid, 7-oxo, Palmitoleic acid, cis-Vaccenic acid, 9-Hexadecenoic acid, cis-10-Nonadecenoic acid, cis-10-Heptadecenoic acid, 9-Eicosenoic acid, (Z) |
| 21. | Fatty alcohols | 1-Dodecanol |
| 22. | Heterocyclic boron-containing compound | 1,3,2-Dioxaborinane, 2,4-diethyl-5-methyl-6-propyl |
| 23. | Heterocyclic organic compounds | Pyrrolo[1,2-a] pyrazine-1,4-dione, hexahydro-3-(2-methylpropyl), Hexahydro-2H-pyrido(1,2-a) pyrazin-3(4H)-one, Octahydro-1H-pyrido(1,2-c) pyrimidin-1-one, 5,6,7,8-Tetrahydrofurazano[4,5-c] azepin-4-one, 3-[(3,3-Dimethyl-1-azetidinyl) imino]-2,2-dimethyl-1-propanol, 3-Furazancarboxamide, 4-(1-aziridinyl), N-Methyl-9-aza-tricyclo [6.2.2.0(2,7)] dodec-2,4,6,11-tetraene-10-one, 1,3-Dioxolane, 2,4,5-trimethyl, 2-t-Butyl-5-propyl- [1,3] dioxolan-4-one, 1,3-Dioxolane, 4,5-dimethyl-2-pentadecyl, Piperidine, 2-(tetrahydro-2-furanyl), Pyrrolidine-5-one, 2-[3-hydroxypropyl], 2-Pyrrolidinemethanol, 1-methyl, 2-t-Butyl-6-chloromethyl- [1,3] dioxan-4-one, 2-Piperazinimine, 1,3,3,4-tetramethyl-N-(methylsulfonyl), 4H-Pyran-4-one, 2,2'-isopropylidenebis [3-methoxy-6-methyl, 5-Isopropenyl-3,3-dimethyl-dihydrofuran-2-one, (2R,5S)-2,6,10,10-Tetramethyl-1-oxaspiro [4.5] decan-7-one, Pyrrolo [1,2-a] pyrazine-1,4-dione, hexahydro-3-(phenylmethyl), Ergotaman-3',6',18-trione, 9,10-dihydro-12'-hydroxy-2'-methyl-5'-(phenylmethyl)-, (5'. alpha.,10. alpha.), 6,7-Dihydro-5H-pyrrolo [2,1-c] [1,2,4] triazole-3-carboxylic acid, 3-Pyrrolidin-2-yl-propionic acid, Piperazine-3,5-dione, 1-tetradecanoyl, 2-Benzimidazolinethione, hexahydro, 2,5-Methanopyrano[3,2-b] pyrrole, hexahydro-1-methyl |
| 24. | Hydrazones | 4-Heptanone, dimethylhydrazone |
| 25. | Imidazoles | 1H-Imidazole, 1-methyl |
| 26. | Ketals | 2-Propanone, 1,1,3,3-tetrabutoxy |

*(Continued)*

**Table 7.** (Continued)

| No. | Compound chemical class | Marine sponge compounds |
|-----|-------------------------|-------------------------|
| 27. | Ketones | 2-Cyclohexen-1-one, 3-methyl, 2-Cyclohexen-1-one, 6-(1-hydroxy-1-methylethyl)-3-methyl |
| 28. | Organic esters | Butanoic acid, 4-chloro-4-oxo-, ethyl ester, Butanoic acid, 2-(hydroxymethyl)-, ethyl ester, (R), Succinic acid, butyl pentyl ester, Butanedioic acid, dibutyl ester, Succinic acid, butyl hexyl ester, 2-Pyrrolidinecarboxylic acid-5-oxo-, ethyl ester, Pivalate, (3-nitrobicyclo [2.2.1] hept-5-en-2-yl) methyl ester, Acetic acid, 2-propylpentyl ester, Acetic acid, chloro-, decyl ester, Cyclohexanecarboxylic acid, 4-propyl-, 4-cyanophenyl ester, trans |
| 29. | Organic nitrogenous compounds | Histamine |
| 30. | Phthalic acid esters | Phthalic acid, 5-methylhex-2-yl pentadecyl ester, Phthalic acid, 5-methylhex-2-yl heptadecyl ester, Phthalic acid, 5-methylhex-2-yl hexadecyl ester, Phthalic acid, butyl undecyl ester, Phthalic acid, 2-ethylhexyl tetradecyl ester |
| 31. | Polycyclic hydrocarbon | [4.2.2] Propella-2,4,7,9-tetraene |
| 32. | Protected sugar derivative | Ribofuranose, 1,5-anhydro-2,3-O-isopropylidene-, d |
| 33. | Pyrazole derivatives | 1H-Pyrazole-1-carboxaldehyde, 4-ethyl-4,5-dihydro-5-propyl |
| 34. | Quinolinedione derivatives | 7-n-Pentadecylaminomethyl-6-hydroxy-5,8-quinolinedione |
| 35. | Spiro compounds | 1-Oxaspiro [4.4] nonan-4-one, 2-isopropyl |
| 36. | Sugar alcohols | 1,3:2,5-Dimethylene-4-methyl-d-rhamnitol, 1,3:2,5-Dimethylene-l-rhamnitol, D-Arabinitol, Xylitol, 6-Desoxy-l-gulitol |
| 37. | Synthetic compounds | Deoxyspergualin |
| 38. | Triazole derivatives | 1-(2-Ethyl-1,2,4-triazol-3-yl) ethanamine |
| 39. | Triglycerides | 9-Octadecenoic acid, 1,2,3-propanetriyl ester, (E, E, E) |

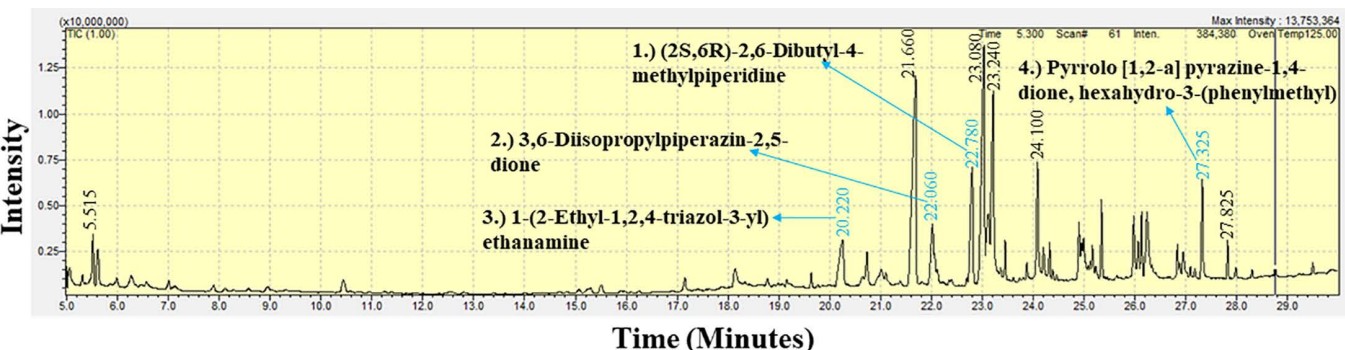

**Fig 6. GC-MS chromatogram analysis of the methanolic extract of *Biemna fistulosa* (BLSi 007), highlighting four potent bioactive compounds.**

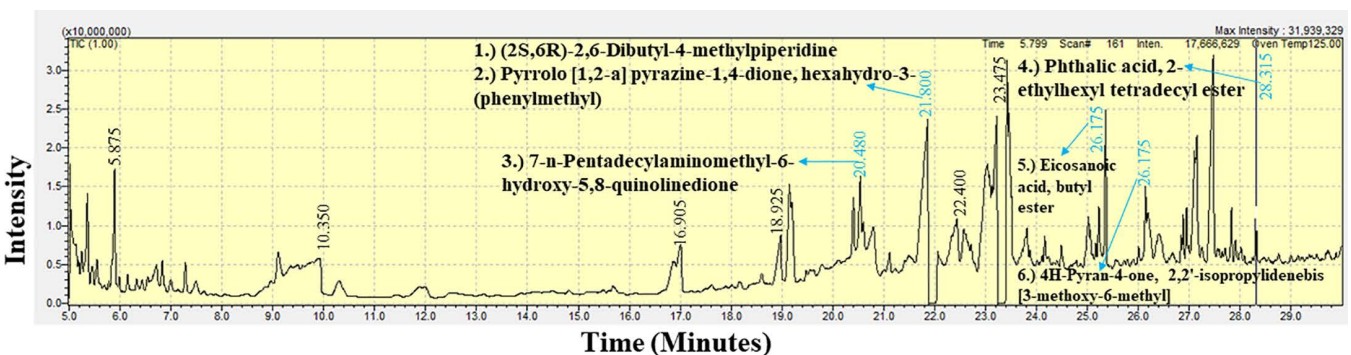

**Fig 7. GC-MS chromatogram analysis of the ethyl acetate extract of *Callyspongia diffusa* (BRMu 004), identifying six secondary bioactive compounds.**

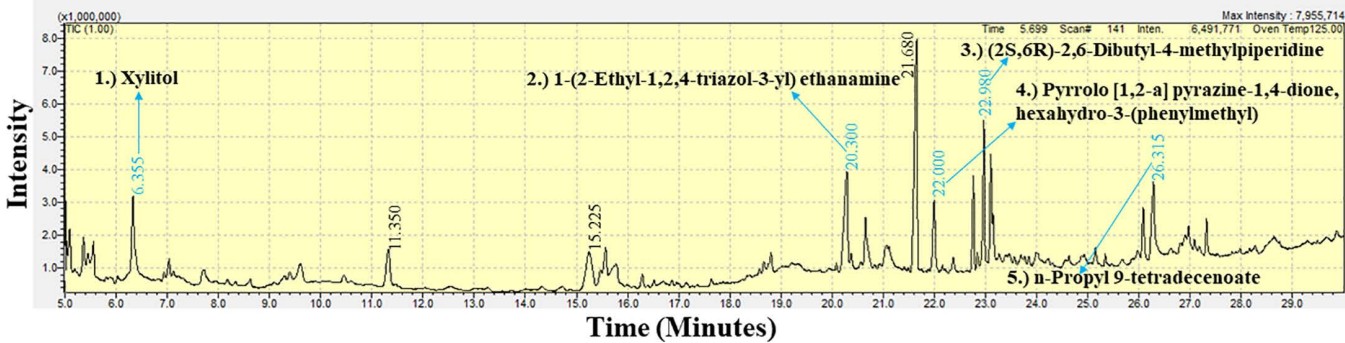

**Fig 8. GC-MS chromatogram analysis of the methanolic extract of *Haliclona fascigera* (BLUCh 014), revealing five bioactive compounds.**

## Discussion

The morphological identification of *B. fistulosa*, *C. diffusa*, and *H. fascigera* offers valuable insights into their taxonomic classification and ecological adaptations. These species exhibit distinct skeletal structures and spicule compositions, which serve as key diagnostic features distinguishing them from closely related taxa [71]. In this study, *B. fistulosa* possessed a fibrous, encrusting growth pattern, characterized by curved diactinal styles and C-shaped sigma microscleres, likely enhancing its ability to anchor onto substrates in dynamic marine environments. Similarly, *C. diffusa* featured a reticulated spongin fiber network interwoven with triangular spicule meshes, contributing to its structural flexibility and resilience in variable hydrodynamic conditions. *H. fascigera*, on the other hand, displayed an isodictyal arrangement of oxeas, raphides, and microstrongyles, a skeletal framework commonly found in sandy lagoon and coral reef habitats, where sediment stabilization and skeletal reinforcement play crucial ecological roles. These skeletal adaptations directly influence the distribution and habitat preferences of these sponges [72].

The encrusting nature of *B. fistulosa* may provide a competitive advantage in wave-exposed environments by reducing drag and facilitating firm attachment to substrates. Similarly, the intricate spicule network in *H. fascigera* supports its survival in sediment-rich coastal habitats, where stability and substrate interaction are essential for persistence. Comparative studies on sponge skeletal morphologies have revealed similar configurations across species belonging to the same genus or geographic region, suggesting potential evolutionary and environmental drivers shaping skeletal development [73].

Beyond their structural attributes, sponge spicules and spongin fibers play essential functional roles in microbial associations and chemical defense. The intricate skeletal matrix of sponges harbors diverse microbial symbionts, with spicules possibly serving as microhabitats for bacteria and other microorganisms [74]. Additionally, bioactive compounds associated with sponge skeletons have demonstrated significant antimicrobial properties, contributing to their defensive strategies against potential pathogenic threats. Research has identified bioactive metabolites within sponge extracts exhibiting antibacterial activity against human pathogens, underscoring the ecological and pharmaceutical relevance of these skeletal structures [75]. A similar study used spicular analysis to investigate the sponge spicule assemblage, including *Biemna, Callyspongia*, and *Haliclona* species, in the lagoon reef of Bocas del Toro, Panama [76]. In Simeulue Island, Aceh Province, Indonesia a morphological study was conducted on over twenty species of marine sponges. Their findings included notable species such as *Carteriospongia foliascens, Biemna fortis, Paratetilla aruensis, Oceanapia sp., Petrosia sp., Haliclona oculata,* and *Haliclona fascigera* [77].

In addition, this study used a DNA barcoding technique and identified three genera (*Biemna, Callyspongia*, and *Haliclona*) within the class Demospongiae. The complementary approaches significantly improved the precision of

**Table 8. Characteristics and antimicrobial activity of selected sponges' natural products identified in the GC-MS analysis of *Biemna fistulosa*, *Callyspongia diffusa*, and *Haliclona fascigera* organic extracts.**

| Sponge extract source | Type of compounds | Retention Time (Minutes) | Compound | Molecular formula | Molecular Weight (g/mol) | Quality of similarity (%) | Bioactivity |
|---|---|---|---|---|---|---|---|
| *Biemna fistulosa, Callyspongia diffusa and Haliclona fascigera* | Cyclic amines | 22.780 | (2S,6R)-2,6-Dibutyl-4-methylpiperidine | $C_{14}H_{29}N$ | 211 | 82 | Antimicrobial activity; inhibits topoisomerase II (DNA gyrase) and topoisomerase IV [83]. |
| | Pyrazole derivative | 22.780 | Pyrrolo[1,2-a]pyrazine-1,4-dione; hexahydro-3-(2-methylpropyl) | $C_{11}H_{18}N_2O_2$ | 210 | 86 | Antifungal [81]; antioxidant and antibacterial activity [82]. |
| | *Biemna fistulosa and Callyspongia diffusa* | | 27.325 | Pyrrolo [1,2-a] pyrazine-1,4-dione, hexahydro-3-(phenylmethyl) | $C_{14}H_{16}N_2O_2$ | 244 | 85 | |
| *Biemna fistulosa, and Haliclona fascigera* | Heterocyclic amine | 20.220 | 1-(2-Ethyl-1,2,4-triazol-3-yl) ethanamine | $C_6H_{12}N_4$ | 140 | 78 | Pharmacological activities such as enzyme inhibition, antifungal, anticancer and antibacterial properties [90]. |
| | L-Proline derivatives | 22.780 | L-Proline, N-valeryl-, heptadecyl ester | $C_{27}H_{51}NO_3$ | 437 | 80 | Antiviral activity in plants [80] and Antitumor agents [81]. |
| *Biemna fistulosa* | Diketopiperazines | 22.060 | 3,6-Diisopropylpiperazin-2,5-dione | $C_{10}H_{18}N_2O_2$ | 198 | 77 | Biological activities, including enzyme inhibition and antimicrobial properties [78]. |
| | | 22.060 | Piperazine-3,5-dione, 1-tetradecanoyl | $C_{18}H_{32}N_2O_3$ | 324 | 76 | |
| | Fatty acids | 27.825 | Decanoic acid, 10-(2-hexylcyclopropyl) | $C_{19}H_{36}O_2$ | 296 | 84 | Used in identifying bacteria and in research for studying its role in bacterial cell membrane protection and metabolism [100]. |
| *Callyspongia diffusa* | Quinolinedione derivative | 20.480 | 7-n-Pentadecylaminomethyl-6-hydroxy-5,8-quinolinedione | $C_{25}H_{38}N_2O_3$ | 414 | 75 | Antibacterial; antifungal; antimalarial; anti-cancer and anti-inflammatory [84]. |
| | Pyrazole derivative | 26.175 | 4H-Pyran-4-one, 2,2'-isopropylidenebis [3-methoxy-6-methyl] | $C_{17}H_{20}O_6$ | 320 | 74 | Antibiotic, anti-inflammatory, antimalarial, antimicrobial, antiviral, anti-diabetic, anti-tumor, herbicidal, insecticidal, and analgesic activities [85]. |
| | Lipophilic chemicals (phthalate acid esters) | 28.315 | Phthalic acid, 2-ethylhexyl tetradecyl ester | $C_{30}H_{50}O_4$ | 474 | 87 | Antimicrobial, insecticidal, allelopathic and other biological activities [86]. |
| | Triglyceride (triolein) | 27.830 | | $C_{57}H_{104}O_6$ | 884 | 83 | Pharmaceuticals as a carrier or excipient in drug formulations; and an emollient and moisturizing therapy in skin care products [88]. |
| *Haliclona fascigera* | Sugar alcohols | 6.355 | Xylitol | $C_5H_{12}O_5$ | 152 | 75 | Antibacterial, diabetes management, ear infection treatment, and gut health [91]; antimicrobial, dental and respiratory health [92]; and anticancer, anti-inflammation and bone health [81]; |
| | Alkyl esters | 26.315 | n-Propyl 9-tetradecenoate | $C_{17}H_{32}O_2$ | 268 | 78 | Antifungal activity against *Candida albican* [97]. |
| | Ester ergot alkaloids | 11.350 | Dodecanoic acid, 2-penten-1-yl ester | | | | Anticancer, antioxidant, and antimicrobial activities [81]. |

taxonomic classification for the marine sponges. The phylogenetic analysis clustered the sponge samples into three sub-clusters, with each representing a distinct genus. The specimen *Callyspongia* sp. BRMu004 (PQ329108) formed a distinct sub-cluster (supported by 100% bootstrap value) with members of the genus *Callyspongia* and had 100% sequence identity with *Callyspongia diffusa* (KX454494) (Fig 5; Table 1). On the other hand, *Haliclona* sp. BLUCh014 (PQ997929) clustered together with members of the genus *Haliclona* that had 99% sequence similarity. The specimen *Biemna* sp. BLSi007 (PQ997931) formed its sub-cluster (supported by 100% bootstrap value) with species from the genus *Biemna.* The formation of separate sub-clusters by the three sponge specimens on the phylogenetic tree indicated that they represented distinct sponge species. The formation of distinct sub-clusters in the phylogenetic tree highlights the genetic divergence among these sponge specimens, confirming their classification as separate species. This molecular approach not only refines taxonomic resolution but also provides insights into evolutionary relationships within *Demospongiae*. The integration of DNA barcoding with phylogenetic analysis is particularly valuable in marine sponge research, where morphological plasticity often complicates species identification. Future studies could expand on this by incorporating additional genetic markers and broader taxonomic sampling to further elucidate phylogenetic relationships and species diversity within these genera [78]. A previous study examined species of shallow Hawaiian sponge fauna in the United States and identified *Biemna fistulosa* and *Callyspongia diffusa* using DNA barcoding [79].

The GC-MS chromatogram data indicated a total of 114 marine sponge chemical compounds from genera *Biemna, Callyspongia,* and *Haliclona*. Notably, this study confirmed that 11.4% of the identified compounds had previously demonstrated antifungal, antibacterial, and antiviral bioactivity. A previous study demonstrated that *Callyspongia* species from the Red Sea exhibited antimicrobial properties, with bioactivity observed in the methanolic extract, its various fractions, and particularly in extracts purified from the dichloromethane fraction [80]. Marine sponges belonging to the genus *Haliclona* are recognized for their capacity to biosynthesize a wide spectrum of secondary bioactive compounds such as sesquiterpenoid quinols, sterols, glycosphingolipids, and bioactive alkaloid compounds [46].

The findings in this study indicate that the ethyl acetate extracts of *C. diffusa* exhibited notable inhibitory activity against *P. aeruginosa*. The GC-MS analysis identified 62 chemical compounds in the ethyl acetate extract of *C. diffusa*, of which 9.7% have been previously reported to exhibit antimicrobial activity. These bioactive compounds include pyrrolo [1,2-a] pyrazine-1,4-dione, hexahydro-3-(phenylmethyl) [81] and [82], (2S,6R)-2,6-Dibutyl-4-methylpiperidine [83], 7-n-Pentadecylaminomethyl-6-hydroxy-5,8-quinolinedione [84], 4H-Pyran-4-one, 2,2'-isopropylidenebis [3-methoxy-6-methyl] [85], phthalic acid, 2-ethylhexyl tetradecyl ester [86] and pyrrolo [1,2-a] pyrazine-1,4-dione; hexahydro-3-(2-methylpropyl)) [81] and [82]. A similar study identified 212 bioactive compounds from the genus *Callyspongia,* and 109 molecules were reported to exhibit bioactivity [87]. Additionally, this research established that the ethyl acetate extract derived from *C. diffusa* demonstrated bactericidal activity against *P. aeruginosa*, with an MBC value of 2.5 mg mL$^{-1}$. Marine sponge extracts from a study of the waters of Mauritius exhibited low MBC values, indicating that these extracts could serve as a potential approach to traditional bacterial infection management strategies [18]. Furthermore, in this study, the 9-octadecenoic acid, 1,2,3-propanetriyl ester E, E, E, a triolein compound, was extracted from the ethyl acetate extract of *C. diffusa*. This compound holds pharmaceutical significance as a carrier or excipient in drug formulations and is also employed as an emollient and moisturizing agent in dermatological products [88]. In a previous study, the bioactive compound 9-Octadecenoic acid, 1,2,3-propanetriyl ester (E, E, E) was discovered in *Cassia angustifolia* and demonstrated significant antimicrobial activity [89]. This underscores the genus's significance as a prolific source of secondary metabolites with promising pharmaceutical applications. The findings of this study are consistent with previous research, reinforcing the role of marine sponges in drug discovery and the development of antimicrobial agents. Further studies should prioritize the isolation and characterization of these bioactive compounds to elucidate their mechanisms of action and explore their potential for therapeutic formulation [87].

Notably, the methanolic extracts from the marine sponges *B. fistulosa* and *H. fascigera* showed significant antibacterial activity against *E. coli*. Among the compounds identified in this study, 4.4% exhibited antibacterial activity, including pyrrolo[1,2-a] pyrazine-1,4-dione, hexahydro-3-(phenylmethyl) [81] and [82], 1-(2-ethyl-1,2,4-triazol-3-yl) ethanamine [90], 7-n-pentadecylaminomethyl-6-hydroxy-5,8-quinolinedione [84], 4H-pyran-4-one, 2,2'-isopropylidenebis [3-methoxy-6-methyl] [85], and xylitol [91] and [92]. In our study, *H. fascigera* exhibited broad-spectrum antibacterial activity, with an inhibition zone diameter (IZD) of $28.33 \pm 3.8$ mm against *E. coli*. Additionally, the *H. fascigera* extract exhibited the highest antibacterial activity among the tested sponge species. It also demonstrated the lowest Minimum Inhibitory Concentration (MIC) against *E. coli* ($0.53 \pm 0.01$ mg mL$^{-1}$), surpassing the efficacy of streptomycin ($1.36 \pm 0.00$ mg mL$^{-1}$). Furthermore, its Minimum Bactericidal Concentration (MBC) was recorded at 1.25 mg/mL, outperforming streptomycin (2.5 mg mL$^{-1}$). These findings confirm that *H. fascigera's* methanolic extract possesses superior antibacterial potential compared to the extracts of *B. fistulosa* and *C. diffusa*, indicating its promising application in antimicrobial drug development. In a similar study, *H. fascigera*, collected from Bidong Island, Malaysia, showed positive results in all in vitro biological studies and exhibited higher antibacterial activities compared to other sponges in the study [93]. Another research also revealed that the butanol fraction of *Biemna* sp. exhibited significant antibacterial activity, with an MIC value of 0.091 mg mL$^{-1}$ against *Escherichia coli* [94].

In a related study, extracts from *H. fascigera* sourced in Indonesia proved effective in inhibiting the growth of *S. aureus* and *E. coli* [95]. Another study from Badi Island of Spermonde Archipelago also revealed the antibacterial activities of crude extracts of *H. fascigera* against three shrimp pathogenic bacteria [48]. In Mauritius, revealed that the organic sponge extracts exhibited greater antibacterial activity than the standard antibiotic against *S. aureus* and *E. coli* [94].

*Candida albicans* is the predominant causative agent of candidiasis, accounting for approximately 70% of fungal infections worldwide and contributing to the annual mortality of over 1.6 million people due to fungal diseases [96]. On one hand, *B. fistulosa* demonstrated more potent bioactive compounds capable of inhibiting C. albicans growth compared to *C. diffusa* and *H. fascigera*; however, its activity did not exceed that of the positive control. Notably, 4.4% of the identified bioactive compounds were previously reported to possess antifungal properties, including pyrrolo [1,2-a] pyrazine-1,4-dione, hexahydro-3-(phenylmethyl) [81] and [82], pyrrolo[1,2-a] pyrazine-1,4-dione; hexahydro-3-(2-methylpropyl) [81] and [82], 1-(2-ethyl-1,2,4-triazol-3-yl) ethanamine [90], 7-n-pentadecylaminomethyl-6-hydroxy-5,8-quinolinedione [84], and n-propyl 9-tetradecenoate [97]. The findings of this study indicate that the methanolic extract of *B. fistulosa* exhibited the most potent fungicidal activity, with a minimum fungicidal concentration (MFC) of 2.5 mg mL$^{-1}$. This suggests that the extracts of *B. fistulosa* are more effective at eliminating *C. albicans* at lower concentrations compared to the other extracts evaluated. Furthermore, research conducted in Biak, Indonesia, demonstrated that the ethyl acetate extracts of *Fascaplysinopsis* sp. and *Haliclona* sp. possess notable antifungal activity against *C. albicans* [98]. Additionally, investigations on marine sponges from the Ratnagiri coast of India reported moderate antifungal activity against *C. albicans* [99]. Moreover, from our study, the decanoic acid, 10-(2-hexylcyclopropyl), a fatty acid obtained from the methanolic extract of *B. fistulosa*, has been used in identifying bacteria and in research for studying its role in bacterial cell membrane protection and metabolism [100].

Xestodecalactones B compounds isolated from *Xestospongia exigua* in the Bali Sea demonstrated antifungal activity against *C. albicans* [46]. Similarly, nortetillapyrone, a tetrahydrofurylhydroxypyran-2-one derived from *Haliclona cymaeformis*, exhibited antifungal efficacy against various fungal pathogens with distinct MIC values [101]. Furthermore, polyketide compounds such as woodylides A and C from *Plakortis simplex*, as well as theonellamide G, swinholide I, and hurghadolide A from *Theonella swinhoei*, along with tetramic acid glycosides aurantosides G and I, demonstrated significant antifungal activity, underscoring the potential of marine sponges as reservoirs of antifungal compounds [102].

## Conclusion and recommendations

This research study underscores the significance of marine sponges as a promising reservoir of antimicrobial agents. Extracts derived from *B. fistulosa*, *C. diffusa*, and *H. fascigera* exhibited pronounced antibacterial activity compared to the positive control. Furthermore, the methanolic extract of *H. fascigera* demonstrated the most potent antibacterial and

antifungal activity among the marine sponge extracts evaluated. The findings of this study suggest that marine sponges from Kenyan waters possess notable therapeutic potential, presenting valuable lead compounds for drug discovery and development. More studies should focus on the mechanisms of action and toxicity of pure leads isolated from sponges at the molecular level, which is important to give direction for lead improvement and further drug development.

## Disclosure

A preprint has been published [103].

## Supporting information

**S1 Table. Summary of marine sponge species observed in high abundance (≥4 sites) across Kenyan coastal study areas.**
(PDF)

**S2 Table. Summary of marine sponge species recorded at moderate abundance (2–3 sites) along the Kenyan coastline.**
(PDF)

**S3 Table. Summary of marine sponge species observed at rare abundance (single site) along the Kenyan coastline.**
(PDF)

## Acknowledgments

We extend our sincere gratitude to Mr. Masudi Juma Zamu, principal diver at the Kenya Marine and Fisheries Research Institute (KMFRI), along with Mr. Kennedy Agoi and Ms. Diana Chepngetich of the Technical University of Mombasa (TUM), for their invaluable technical support during both fieldwork and laboratory activities. Their professionalism and dedication greatly enhanced the success of this study.

## Author contributions

**Conceptualization:** Teresia Nyambura Wacira, Huxley Mae Makonde, Joseph Nyingi Kamau, Christopher Mulanda Aura, Cromwell Mwiti Kibiti.

**Data curation:** Teresia Nyambura Wacira, Huxley Mae Makonde, Joseph Nyingi Kamau.

**Formal analysis:** Teresia Nyambura Wacira.

**Funding acquisition:** Teresia Nyambura Wacira, Joseph Nyingi Kamau.

**Investigation:** Teresia Nyambura Wacira, Huxley Mae Makonde, Cromwell Mwiti Kibiti.

**Methodology:** Teresia Nyambura Wacira, Cromwell Mwiti Kibiti.

**Project administration:** Teresia Nyambura Wacira, Joseph Nyingi Kamau, Cromwell Mwiti Kibiti.

**Resources:** Teresia Nyambura Wacira, Joseph Nyingi Kamau.

**Software:** Teresia Nyambura Wacira.

**Supervision:** Huxley Mae Makonde, Cromwell Mwiti Kibiti.

**Validation:** Huxley Mae Makonde, Cromwell Mwiti Kibiti.

**Visualization:** Teresia Nyambura Wacira.

**Writing – original draft:** Teresia Nyambura Wacira, Cromwell Mwiti Kibiti.

**Writing – review & editing:** Huxley Mae Makonde, Joseph Nyingi Kamau, Christopher Mulanda Aura, Cromwell Mwiti Kibiti.

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
