## [Decision Letter · Decision Letter 0]

Dear Dr. WACIRA,

Thank you for submitting your manuscript to PLOS ONE. After careful consideration, we feel that it has merit but does not fully meet PLOS ONE’s publication criteria as it currently stands. Therefore, we invite you to submit a revised version of the manuscript that addresses the points raised during the review process.

We look forward to receiving your revised manuscript.

Kind regards,

Awatif Abid Al-Judaibi, PhD

Academic Editor

PLOS ONE

“We sincerely thank the Kenya Marine and Fisheries Research Institute (KMFRI) for their invaluable support in partially funding for the research  (GOK-PC Target C82 39-1).”Please state what role the funders took in the study.  If the funders had no role, please state: "The funders had no role in study design, data collection and analysis, decision to publish, or preparation of the manuscript."

4. Thank you for stating the following in the Funding Section of your manuscript:

“We sincerely thank the Kenya Marine and Fisheries Research Institute (KMFRI) for their invaluable support (GOK-PC Target C82 39-1).”

“We sincerely thank the Kenya Marine and Fisheries Research Institute (KMFRI) for their invaluable support in partially funding for the research  (GOK-PC Target C82 39-1).”Please state what role the funders took in the study.  If the funders had no role, please state: "The funders had no role in study design, data collection and analysis, decision to publish, or preparation of the manuscript."

5. We note that Figures 1, 2, 3, and 4 in your submission contain copyrighted images. All PLOS content is published under the Creative Commons Attribution License (CC BY 4.0), which means that the manuscript, images, and Supporting Information files will be freely available online, and any third party is permitted to access, download, copy, distribute, and use these materials in any way, even commercially, with proper attribution. For more information, see our copyright guidelines: http://journals.plos.org/plosone/s/licenses-and-copyright.

1. You may seek permission from the original copyright holder of Figures 1, 2, 3, and 4 to publish the content specifically under the CC BY 4.0 license.

Reviewers' comments:

Reviewer's Responses to Questions

**Comments to the Author**

1. Is the manuscript technically sound, and do the data support the conclusions?

Reviewer #1: Yes

Reviewer #2: Yes

Reviewer #3: Partly

2. Has the statistical analysis been performed appropriately and rigorously?

Reviewer #1: Yes

Reviewer #2: Yes

Reviewer #3: I Don't Know

3. Have the authors made all data underlying the findings in their manuscript fully available?

Reviewer #1: Yes

Reviewer #2: Yes

Reviewer #3: Yes

4. Is the manuscript presented in an intelligible fashion and written in standard English?

Reviewer #1: Yes

Reviewer #2: Yes

Reviewer #3: Yes

Reviewer #1: - In the financial disclosure, there's missing information, like: Who received the funding? and if the sponsors or funders play any role in this study?.

- Line 13 - 15: It's not necessary to rewrite the name in "Corresponding Author"; simply enter an asterisk (*) followed by the email address. Please review PLOS Affiliations Guidelines.

- Line 17 - 20: Lots of information about the authors, I believe that data is stored internally when the article is submitted. Please review PLOS Affiliations Guidelines.

-Line 164 - 166: Important to indicate the conditions under which the purified products were shipped.

Reviewer #2: All my comments have been included in the reviewed document. I suggest including ethical considerations in the manuscript and the permissions for collecting the sponges.The revision of each part of the document is indicated in the manuscript.

Reviewer #3: 1. Summary of the research

Wacira et al. used morphological and genetic analyses to identify three sponge species and tested their extracts against human pathogens E. coli, P. aeruginosa, S. aureus, and C. albicans. In particular, the authors measured inhibition zone diameters and determined the minimum inhibitory concentrations for each. Furthermore, the authors identified the chemical compounds present. The authors conclude that all three species of sponge showed antimicrobial activity against at least one of the pathogens. Specifically, methanolic extracts from B. fistulosa and H. fascigera had larger inhibition zone diameters for E. coli than the positive control, while ethyl acetate extracts of C. diffusa had a larger inhibition zone diameter for P. aeruginosa than the positive control. While methanolic extracts from B. fistulosa had the largest inhibition zone diameter of all treatments for C. albicans, none were greater than the positive control. H. fascigera had the lowest MBC against E. coli, even lower than the positive control. This study addresses an important topic and provides valuable insights into bioactivity potential of marine sponges. However, I believe rearranging the structure of how the results are presented will improve readability, and expanding on certain details in the introduction and discussion would be necessary to provide adequate context for the study. I recommend the below changes.

2. Major issues

(1) Further context needed in introduction

• Line 82: What are the gaps in knowledge? This paragraph could be expanded i.e. discuss if there are challenges with identifying sponges with morphological techniques, advantages/disadvantages of extract types (methanolic, ethyl acetate..), is there anything else unknown about chemical compounds? If the gap is mainly from the geographic location, perhaps consider including further site details such as oceanography or current bioactive compounds work/industry from the location. I thought the paragraph beginning on line 66 provided good context with examples of chemical compounds with recorded bioactive properties

(2) Results were somewhat hard to follow starting with section “In Vitro antibiotic and antifungal activity of the marine sponge crude extracts”

• Since the text generally discusses results by pathogen and this appears to be the main focus of the manuscript, I suggest to reorganize Tables 2-4 so that each pathogen is one table with the extract types by column and sponges still by row. I believe this will be easier for the reader to see the main points

• I believe the paragraph beginning on line 419 is repeated several times. I think it should either start on line 403, before "All three [..]" or go in the table captions

• Lines 455-469 are especially hard to follow. Line 455: Streptomycin exhibited lowest compared to what? Does the next sentence contradict this? Line 459: what reference drugs does this refer to? Where are these values from, are citations needed here?

(3) Discussion needs a bit more context and seems to jump around a bit starting with line 634.

• Good context is provided by discussing examples of chemical compounds with bioactivity of these sponges, however further context is needed in some places i.e. lines 556, 574, and 598

• After discussing morphological and genetic identification, consider ordering paragraphs by result combination rather than methods. I.e. so that methyl acetate extracts of C. diffusa exhibited inhibitory activity against P. aeruginosa is discussed, followed by identified chemical compounds for this, followed by MIC. Then discuss methanolic extracts from B. fistulosa and H. fascigera against E. coli in the same format, etc. (Suggest to move paragraph beginning on line 634 to 613, and paragraph beginning on line 644 to after that one)

• Line 660: is H. fascigera meant to be B. fistulosa? I think that is what the results and tables state. This paragraph seems most related to paragraph 614 and should be moved to line 633

• Line 661: I am not clear on where this value is from?

3. Minor issues

(1) Line 49: I believe the first two sentences were meant to be one sentence separated with a comma

(2) Lines 55 and 56: “class” and “order” I think should be capitalized

(3) Line 93: suggest to state “three” species of sponge instead of “some”

(4) Line 262: suggest to add “and fungal” between “bacterial growth” for clarity

(5) Line 266: For clarity suggest to add that MBC and MFC were determined for those pathogens that showed an inhibitory zone, if that is indeed the case

(6) Line 302: were the data tested for normal distribution? What were the sample sizes? Also suggest to specify the factors in the ANOVA for clarity

(7) Line 319: how long were samples out of water for? Could this have impacted results?

(8) Line 558: delete “conducted”

(9) Line 563: delete comma

(10) Line 578: delete “t” in “indicated”

(11) Line 609: provide geographic context of study discussed

(12) Line 614: I thought this was good context and the previous paragraph discussing the bacterial pathogens could benefit from a similar level of detail

(13) Line 616: add that it was not higher than the positive control here

4. Other points (confidential)

I have selected 'yes' in #3 that the authors have provided their data. However I am not sure because the tables in the manuscript only contain summary statistics. I am available to review a revised version.

**Do you want your identity to be public for this peer review?** For information about this choice, including consent withdrawal, please see our Privacy Policy

Reviewer #1: No

Reviewer #2: **Yes: ** José Iannacone

Reviewer #3: No

---

## [Author Response · Author response to Decision Letter 1]

23 Jun 2025

Response to reviewers

Dear Dr. Awatif Abid Al-Judaibi,

Thank you for the opportunity to revise our manuscript. We have carefully addressed all comments from the Academic Editor and reviewers, as detailed in the attached ‘Response to Reviewers’ document. Key revisions include enhanced methodological clarity, improved interpretation of results, and refined formatting for better readability.

Regarding Figures 1–4, we confirm that all images were originally generated by the corresponding author using in-house microscopy equipment. No third-party content was used. To ensure transparency, captions have been updated to include “(Source: Author)” and descriptive metadata has been added in Lines 347–396. Should any verification be required, we are happy to provide supporting documentation.

We sincerely appreciate your guidance throughout the revision process and trust that the updated manuscript meets the journal’s standards.

Dear Reviewer #1,

Thank you for your helpful comments. We have revised the manuscript accordingly. The financial disclosure now specifies that the corresponding author received funding for sponge sampling and DNA sequencing, and we’ve added the standard statement that funders had no role in the research process. Author information has been updated to meet PLOS formatting guidelines, including a correction to the corresponding author line and removal of redundant contact details. Additionally, we included the shipping conditions for purified products, specifying that samples were preserved under dry ice and sent to Inqaba Biotec for sequencing. We appreciate your input in strengthening the manuscript.

Dear Reviewer #2,

Thank you for your thoughtful and constructive comments. We have addressed all points in the revised manuscript. Ethical approvals and field collection permits have been clearly stated, and corresponding documentation will be provided as supplementary files. Clarifications have been added regarding the most active sponge species (H. fascigera) against tested pathogens, including S. aureus and C. albicans, with supportive data cited from the results and discussion sections. The introduction now includes a corrected reference and additional detail on Kenyan sponge diversity. We have refined the study aim for specificity, corrected the taxonomic reference to B. fistulosa, and repositioned methods and analytical details for better alignment. Minor textual refinements were also made to improve clarity and coherence. We trust these revisions meet the journal’s expectations and thank you again for your guidance.

Dear Reviewer #3,

Thank you for your thoughtful and constructive comments. We have revised the manuscript to address all points raised. Specifically, we clarified ambiguous comparisons related to streptomycin MIC and MBC values, expanded the discussion with additional ecological and phylogenetic context, corrected a taxonomic mislabelling, and improved sentence structure and terminology throughout. Methodological descriptions, including statistical analysis and sample handling, were also refined for accuracy. Minor textual and formatting corrections were made to ensure consistency. These revisions have strengthened the clarity, coherence, and scientific rigor of the manuscript.

Kind regards,

Name: Ms. TERESIA NYAMBURA WACIRA

E-mail: tnyambura@kmfri.go.ke

---

## [Decision Letter · Decision Letter 1]

Characterization and Bioactivity Potential of Marine Sponges (Biemna fistulosa, Callyspongia diffusa, and Haliclona fascigera) from Kenyan Coastal Waters

PONE-D-25-26894R1

Dear Dr. TERESIA NYAMBURA WACIRA,

We’re pleased to inform you that your manuscript has been judged scientifically suitable for publication and will be formally accepted for publication once it meets all outstanding technical requirements.

Kind regards,

Awatif Abid Al-Judaibi, PhD

Academic Editor

PLOS ONE

Reviewers' comments:

Reviewer's Responses to Questions

**Comments to the Author**

Reviewer #2: All comments have been addressed

Reviewer #3: All comments have been addressed

2. Is the manuscript technically sound, and do the data support the conclusions?

Reviewer #2: Yes

Reviewer #3: Yes

3. Has the statistical analysis been performed appropriately and rigorously?

Reviewer #2: Yes

Reviewer #3: Yes

4. Have the authors made all data underlying the findings in their manuscript fully available?

Reviewer #2: Yes

Reviewer #3: Yes

5. Is the manuscript presented in an intelligible fashion and written in standard English?

Reviewer #2: Yes

Reviewer #3: Yes

Reviewer #2: (No Response)

Reviewer #3: (No Response)

**Do you want your identity to be public for this peer review?** For information about this choice, including consent withdrawal, please see our Privacy Policy

Reviewer #2: **Yes: ** José Iannacone

Reviewer #3: No

---

## [Editor Report · Acceptance letter]

PONE-D-25-26894R1

PLOS ONE

Dear Dr. Wacira,

I'm pleased to inform you that your manuscript has been deemed suitable for publication in PLOS ONE. Congratulations! Your manuscript is now being handed over to our production team.

Kind regards,

on behalf of

Professor Awatif Abid Al-Judaibi

Academic Editor

PLOS ONE